# Bayesian Causal Bandits with Backdoor Adjustment Prior

**Jireh Huang**                                                                           *jirehhuang@ucla.edu*
*Department of Statistics*
*University of California, Los Angeles*

**Qing Zhou**                                                                                 *zhou@stat.ucla.edu*
*Department of Statistics*
*University of California, Los Angeles*

**Reviewed on OpenReview:** *https: // openreview. net/ forum? id= sMsGv5Kfm3*

## Abstract

The causal bandit problem setting is a sequential decision-making framework where actions of interest correspond to interventions on variables in a system assumed to be governed by a causal model. The underlying causality may be exploited when investigating actions in the interest of optimizing the yield of the reward variable. Most existing approaches assume prior knowledge of the underlying causal graph, which is in practice restrictive and often unrealistic. In this paper, we develop a novel Bayesian framework for tackling causal bandit problems that does not rely on possession of the causal graph, but rather simultaneously learns the causal graph while exploiting causal inferences to optimize the reward. Our methods efficiently utilize joint inferences from interventional and observational data in a unified Bayesian model constructed with intervention calculus and causal graph learning. For the implementation of our proposed methodology in the discrete distributional setting, we derive an approximation of the sampling variance of the backdoor adjustment estimator. In the Gaussian setting, we characterize the interventional variance with intervention calculus and propose a simple graphical criterion to share information between arms. We validate our proposed methodology in an extensive empirical study, demonstrating compelling cumulative regret performance against state-of-the-art standard algorithms as well as optimistic implementations of their causal variants that assume strong prior knowledge of the causal structure.

## 1 Introduction

The multi-armed bandit (MAB) problem is a well-known sequential allocation framework for experimental investigations (Berry & Fristedt, 1985). Classically, the MAB problem formulation features an action set $\mathcal{A}$ consisting of $|\mathcal{A}| = K$ actions, also called arms, typically corresponding to interventions. Each arm $a \in \mathcal{A}$ defines a real-valued distribution for the reward signal, with expected reward $\mu_a$. The objective of an allocation policy is to sequentially pick arms in a manner that maintains a balance between exploration and exploitation in the interest of identifying and obtaining the greatest reward. Maximally and effectively utilizing all available information is imperative, especially when investigating interventions that are either or both resource-demanding and time-consuming.

Lattimore et al. (2016) proposed the causal bandit (CB) problem setting wherein a non-trivial probabilistic causal model is assumed to govern the distribution of the reward variable and its covariates (Pearl, 2000). The addition of causal assumptions introduces avenues by which interventional distributions may be inferred from observational distributions and information may be shared between arms. Most works addressing the CB problem exploit strong assumptions as to prior knowledge of the underlying causal model to achieve improvements over standard MAB algorithms. In this work, we develop a Bayesian CB framework that

does not require prior knowledge of the underlying causal structure, but instead efficiently utilizes previously available observational data and acquired interventional data to inform exploitation and guide exploration.

For illustrative purposes, we borrow and adapt the farming example described in Lattimore et al. (2016) as an illuminating motivating example of the problem setting of interest and the surrounding challenges. Suppose a farmer wishes to optimize the yield of a certain crop, which she knows is only dependent on temperature, a particular soil nutrient, and moisture level. While she understands that crop yield is somehow affected by these factors, the underlying causality governing this system of four variables (including crop yield) is unknown to her. The farmer's resource limitations restrict her to intervening on at most one factor in each crop season by adjusting the temperature, controlling the soil nutrient content, or regulating moisture level. These experimental interventions are costly to perform, and each realization of the interventional data can only be observed once a season. Hence, it is in the farmer's best interest to leverage her historical logs containing observational data accrued from previous seasons where no interventions were performed, but rather the variables were passively observed as they naturally varied from season to season. In this paper, we propose a framework by which the farmer may aggregate and synthesize the available evidence to optimize resource allocation to attain the highest yield.

## 1.1 Related Work

In its original formulation by Lattimore et al. (2016), the CB problem presupposes knowledge of the underlying causal graph. Accordingly, most proposed CB algorithms require knowledge of the causal graph structure (Lattimore et al., 2016; Lee & Bareinboim, 2018; Maiti et al., 2021; Yabe et al., 2018), and some additionally assume certain model parameters are given (Lu et al., 2020; Nair et al., 2021). Furthermore, many approaches are dependent on some restrictive form or class of graphs. These assumptions are restrictive and often unrealistic in practice.

More recently, Lu et al. (2021) proposed a central node approach based on the work of Greenewald et al. (2019) that does not assume prior knowledge of the causal graph, but rather asymptotic knowledge of the observational distribution. Their approach is restrictive in terms of structural and distributional assumptions, and while it is generally reasonable to assume that observational data is much more accessible than interventional data (Greenewald et al., 2019), the large-sample observational setting is not often realistic. de Kroon et al. (2022) proposed an estimator using separating sets to share information between arms without assuming prior knowledge of nor requiring discovery of the causal graph. Their methodology makes no attempt to learn the causal graph, and makes use of observational data only to strengthen conditional independence testing to identify separating sets.

Relevant to our work is intervention calculus, a set of inference rules proposed by Pearl (2000) that defines avenues by which interventional probabilities may be estimated from observational data. In the CB setting, Lattimore et al. (2016) and Nair et al. (2021) consider graph structures with no confounding such that the interventional distributions are equivalent to conditional distributions, and Maiti et al. (2021) proposed a consistent estimator for the expected reward for discrete variables using both interventional and observational data in the presence of confounding. Our work extends the Bayesian model averaging approach proposed by Pensar et al. (2020) wherein the possible causal effect estimates are averaged across an observational posterior distribution of graphs.

Bayesian approaches to the MAB problem has received significant attention in the past decade. Russo & Van Roy (2014) translated existing regret bounds of algorithms based on optimism in the face of uncertainty to Bayesian regret bounds for posterior sampling by establishing a deep connection between the two in their study of the Bayesian regret. In their information-theoretic analysis of Thompson sampling, Lu & Van Roy (2019) demonstrated improved regret performance by leveraging prior information regarding the bandit environment, which was further tightened in the form of a prior-dependent bound by Kveton et al. (2021). Numerous developments to learn the prior have been proposed in the bandit meta-learning space (Basu et al., 2021; Kveton et al., 2021; Wan et al., 2021; Hong et al., 2022), with many works studying the effects of prior misspecification (Bastani et al., 2022; Peleg et al., 2022; Simchowitz et al., 2021). While these contributions operate in the multi-task setting, seeking to learn the prior by solving many similar tasks typically corresponding to bandit instances, our work more resembles the approach of Kaufmann et al. (2012)

in applying Bayesian techniques to tackle a single bandit instance. In such a way, rather than an estimate of the task prior, our proposed prior is best interpreted as encoding the prior information on the underlying parameters of a given bandit instance based on available observational data generated within the task.

## 1.2 Our Contributions

We approach the CB problem from a Bayesian perspective, assuming simply that finite samples of observational data are available. Importantly, we do not assume the causal graph is known, nor are we restrictive as to the class of graph structures. We design a novel Bayesian CB framework called **B**ayesian **B**ackdoor **B**andit (BBB) that efficiently utilizes the entirety of evidence from an ensemble of observational and interventional data in a unified Bayesian model. Our proposed BBB methodology quantifies the uncertainty in the expected reward estimates as contributed to by the reward signal and the causal model to identify potentially profitable exploration, simultaneously learning the causal graph in addition to and for the purposes of improving estimates to exploit. Through extensive numerical experiments, we validate our methodology by demonstrating compelling empirical performance against both non-causal and causal algorithms. In particular, we show that our BBB approach is able to leverage modest samples of observational data, seamlessly integrating causal effect estimation and structure modeling, to achieve substantially superior cumulative regret performance compared to standard algorithms that make no use of observational information. We similarly demonstrate competitive performance against a generously optimistic version of the causal central node approach proposed by Lu et al. (2021) that assumes large-sample observational data. A preliminary analysis shows that, under some assumptions on the posterior distributions in our Bayesian model, the dependence of the cumulative regret of a BBB algorithm on the size of the action space can be greatly relaxed given sufficient amount of observational data.

Additionally, in detailing the application of our methods to the discrete and Gaussian distributional settings, we propose various developments that are of independent interest. In the discrete setting, we derive an approximation for the sampling variance of the backdoor adjustment probability estimate. In the Gaussian setting, we characterize the interventional variance of a target variable using intervention calculus and correspondingly propose an estimator, and we propose a simple graphical criterion for sharing causal information between arms to perform intervention calculus with jointly observational and interventional data.

The remainder of the paper is arranged as follows. We first review relevant background and notation in Section 2. Then, we develop the formulation of our proposed Bayesian backdoor adjustment prior and its posterior update in Section 3, discussing the design of informative conditional priors given a graph and Bayesian model averaging across graph structures. In Section 4, we develop our proposed algorithms by applying established MAB algorithms under the BBB framework, and we discuss details regarding the implementation of BBB in the discrete and Gaussian settings in Section 5. Finally, we provide extensive empirical results in Section 6 and conclude with a discussion in Section 7. Appendices A through D contain proofs, additional details and numerical results, and technical derivations.

## 2 Preliminaries

We consider the setting where the generative model governing a joint probability distribution $P$ of a set of $p$ variables $\mathbf{X} = \{X_1, \ldots, X_p\}$ is a causal Bayesian network (CBN). A CBN model is defined by $\mathcal{B} = (\mathcal{G}, \Theta_{\mathcal{G}})$ consisting of its structure $\mathcal{G}$, which takes the form of a directed acyclic graph (DAG), and its parameters $\Theta_{\mathcal{G}}$. Its DAG $\mathcal{G} = (\mathbf{V}, \mathbf{E})$, often referred to as the underlying causal graph, is composed of a set of nodes $\mathbf{V} = \{1, \ldots, p\}$ in one-to-one correspondence with the variables, and a set of directed edges $\mathbf{E}$ oriented such that there are no directed cycles. As is standard in causal literature, we may refer to a node $i \in \mathbf{V}$ and its corresponding variable $X_i \in \mathbf{X}$ interchangeably. In our work, we assume that $\mathbf{X}$ is a causally sufficient system with no unobserved confounders.

The causal implications imposed by the underlying CBN of $\mathbf{X}$ are expressed in the form of a structural equation model (SEM), $X_i = f(\mathbf{Pa}_i^{\mathcal{G}}, \varepsilon_i)$ for all $i \in \mathbf{V}$, where $\mathbf{Pa}_i^{\mathcal{G}} = \{X_j : j \rightarrow i \in \mathbf{E}\}$ is the parents of $X_i$ in $\mathcal{G}$ and $\varepsilon_i$ is an exogenous noise term. Otherwise stated, each variable $X_i$ is a function of its direct causes in $\mathcal{G}$ and an independent noise variable, which defines its conditional distribution $P(X_i \mid \mathbf{Pa}_i^{\mathcal{G}}, \theta_i^{\mathcal{G}})$

with local parameters $\theta_i^{\mathcal{G}} \in \Theta_{\mathcal{G}}$. The joint distribution $P$ imposed by the CBN factorizes according to structure $\mathcal{G}$: $P(\mathbf{X}) = \prod_{i=1}^{p} P(X_i \mid \mathbf{Pa}_i^{\mathcal{G}}, \theta_i^{\mathcal{G}})$. Realizations from $P(\mathbf{X})$ without any experimental intervention are referred to as observational data, whereas interventional data are realizations of the SEM when the value of one or more variables are being controlled by intervention. In our work, we consider deterministic atomic interventions denoted $do(X_j = x_j)$ (Pearl, 1995), where a single variable is forcibly controlled to a fixed value $x_j \in \text{Dom}(X_j)$. This has the effect of mutilating the causal graph by deleting the direct effects of $\mathbf{Pa}_j^{\mathcal{G}}$ on $j$, correspondingly modifying the SEM to $X_j = x_j$ and $X_i = f(\mathbf{Pa}_i^{\mathcal{G}}, \varepsilon_i)$ for all $i \in \mathbf{V} \setminus j$. The interventional distribution $P(\mathbf{X} \setminus X_j \mid do(X_j = x_j))$ is not in general equivalent to the observational conditional distribution $P(\mathbf{X} \setminus X_j \mid X_j = x_j)$, motivating the calculation of the interventional distribution from the observational distribution, such as (1) below, which we refer to as interventional calculus.

The action set $\mathcal{A}$ consists of $|\mathcal{A}| = K$ arms that correspond to interventions on variables in $\mathbf{X} \setminus Y$, where $Y = X_p$ is the reward variable (Lattimore et al., 2016). In particular, let arm $a \in \mathcal{A}$ correspond to the intervention $do(X_{\langle a \rangle} = x_a)$, fixing $X_{\langle a \rangle}$ to some value $x_a \in \text{Dom}(X_{\langle a \rangle})$, where $\langle a \rangle \in \mathbf{V}$ is the node corresponding to the intervened variable. The expected reward of each arm $a \in \mathcal{A}$ is given by $\mu_a := \mathrm{E}_P[Y \mid do(X_{\langle a \rangle} = x_a)]$ where $\mathrm{E}_P[\cdot]$ is the expectation in $P$ defined by CBN $\mathcal{B}$, and there is some optimal arm $a^* := \text{argmax}_{a \in \mathcal{A}} \mu_a$ corresponding to the optimal reward $\mu^* := \mu_{a^*}$. Given a horizon of time steps $T$, let $a_t \in \mathcal{A}$ be the arm pulled by an algorithm at time step $t \in \{1, \ldots, T\}$. The objective of the algorithm is to pull arms over $T$ time steps with a balance between exploring different arms and exploiting the reward signal to minimize the expected cumulative regret $\mathrm{E}[\sum_{t=1}^{T} (\mu^* - \mu_{a^t})]$.

We now describe a Bayesian approach to the general MAB problem, with some notation adapted from Kaufmann et al. (2012). The parameters $\Theta_{\mathcal{A}} = (\theta_a)_{a \in \mathcal{A}}$, assumed to mutually independently define the corresponding marginal reward distributions $p_{\theta_a}(y) := P[Y = y \mid do(X_{\langle a \rangle} = x_a)]$, jointly follow a modular prior distribution $\Pi^0(\Theta_{\mathcal{A}}) = \prod_{a \in \mathcal{A}} \pi_a^0(\theta_a)$. Typically, $(\pi_a^0)_{a \in \mathcal{A}}$ are chosen to be all equal and uninformative. When arm $a_t \in \mathcal{A}$ is pulled at time step $t$ and a realization $y_t \leftarrow Y \mid do(X_{\langle a_t \rangle} = x_{a_t})$ is observed, the posterior $\Pi^t$ is computed by updating according to $\pi_{a_t}^t(\theta_{a_t}) \propto p_{\theta_{a_t}}(y_t) \pi_{a_t}^{t-1}(\theta_{a_t})$, while $\pi_a^t = \pi_a^{t-1}$ for $a \neq a_t$. For each arm $a \in \mathcal{A}$, the posterior $\pi_a^t$ induces a posterior distribution for the expected reward $\mu_a$, which is simply a marginal or transformation of $\pi_a^t$ since, in general, $\mu_a$ is a function of $\theta_a$. These posteriors are utilized by Bayesian MAB algorithms, which we discuss and apply under our proposed framework in Section 4.

In our problem formulation, we assume possession of $n_0$ samples of observational data $\mathcal{D}_0$ prior to investigating arms. We denote by $\mathcal{D}^{(t)}$ the interventional data acquired by pulling arm $a_t$ at time $t$, and by $\mathcal{D}_a[t] = \bigcup_{l \le t, a_l = a} \mathcal{D}^{(l)}$ the accumulated interventional data from arm $a$ through time $t$. The combined observational and interventional data accrued through time $t$ is $\mathcal{D}[t] = \mathcal{D}_0 \cup \bigcup_{a \in \mathcal{A}} \mathcal{D}_a[t]$, which we refer to as ensemble data.

## 3   Designing Informative Priors with Intervention Calculus

In this section, we detail the design of the cornerstone of BBB and what we refer to as the backdoor adjustment prior, an informative prior $\Pi^0$ that distinguishes BBB from standard non-causal bandit algorithms by encoding inferences from observational data and seamlessly integrating interventional data to update to the posterior $\Pi^t$. We begin by introducing the construction of conditional priors given backdoor adjustment sets before continuing to obtain the backdoor adjustment prior by Bayesian model averaging over parent set probabilities. We conclude by discussing the formulation and considerations of the posterior distribution over graph structures that gives rise to the parent set probabilities.

**Conditional Priors**. For each arm $a \in \mathcal{A}$, we construct conditional priors $\pi_{a|\mathbf{Z}}^0(\theta_a)$ using the backdoor adjustment given sets $\mathbf{Z} \subseteq \mathbf{X} \setminus X_{\langle a \rangle}$ as follows. If $\mathbf{Z}$ satisfies the backdoor criterion relative to $X_{\langle a \rangle}$ and $Y$ (Pearl, 2000, Definition 3.3.1), then the interventional distribution $Y \mid do(X_{\langle a \rangle} = x_a)$ may be expressed in terms of the joint observational distribution of $\{X_{\langle a \rangle}, Y\} \cup \mathbf{Z}$ via the backdoor adjustment (Pearl, 2000, Theorem 3.3.2):

$$P[Y = y \mid do(X_{\langle a \rangle} = x_a)] = \sum_{\mathbf{z} \in \text{Dom}(\mathbf{Z})} P(Y = y \mid X_{\langle a \rangle} = x_a, \mathbf{Z} = \mathbf{z}) P(\mathbf{Z} = \mathbf{z}). \tag{1}$$

Eq. (1) provides an avenue through which an estimator for $\mu_a$ using observational data may be derived, which we denote $\hat{\mu}_{a,\mathrm{bda}}(\mathbf{Z})$ and with which we design an informative prior $\pi^0_{a|\mathbf{Z}}$ such that the induced prior distribution of the expected reward $\mu_a$ satisfies

$$\mathrm{E}_{\pi^0_{a|\mathbf{Z}}}[\mu_a] = \hat{\mu}_{a,\mathrm{bda}}(\mathbf{Z}), \quad \mathrm{Var}_{\pi^0_{a|\mathbf{Z}}}[\mu_a] = \hat{\mathrm{SE}}^2[\hat{\mu}_{a,\mathrm{bda}}(\mathbf{Z})]. \tag{2}$$

Here, $\mathrm{E}_{\pi^0_{a|\mathbf{Z}}}[\cdot]$ and $\mathrm{Var}_{\pi^0_{a|\mathbf{Z}}}[\cdot]$ are respectively the expectation and variance in $\pi^0_{a|\mathbf{Z}}$, and $\hat{\mathrm{SE}}^2[\hat{\mu}_{a,\mathrm{bda}}(\mathbf{Z})]$ is the estimated sampling variability of the expected reward estimate $\hat{\mu}_{a,\mathrm{bda}}(\mathbf{Z})$ in $P$. The matching of the prior variance with the sampling variance of the backdoor adjustment estimator endeavors to assign the appropriate prior effective sample size. When arm $a_t = a$ is pulled at time step $t \in \{1, \ldots, T\}$ and a realization of the reward $y_t \leftarrow Y \mid do(X_{\langle a \rangle} = x_a)$ is observed in $\mathcal{D}^{(t)}$, the posterior $\pi^t_{a|\mathbf{Z}}$ is computed by updating according to $\pi^t_{a|\mathbf{Z}}(\theta_a) \propto p_{\theta_a}(y_t) \, \pi^{t-1}_{a|\mathbf{Z}}(\theta_a)$.

**Parent Set Averaging**. Thus far we have taken for granted the possession of adjustment set $\mathbf{Z}$, the validity of which is dependent on the underlying causal structure $\mathcal{G}$ which we assume to be unknown. If $Y \notin \mathbf{Pa}^{\mathcal{G}}_{\langle a \rangle}$, then $\mathbf{Z} = \mathbf{Pa}^{\mathcal{G}}_{\langle a \rangle}$ satisfies the backdoor criterion relative to $X_{\langle a \rangle}$ and $Y$, and its uncertainty is quantified by the posterior probability $P(\mathbf{Pa}_{\langle a \rangle} = \mathbf{Z} \mid \mathcal{D}[t])$ given the ensemble data at time $t$. Accordingly, the posterior of $\theta_a$ is determined by averaging over all possible parent sets for $X_{\langle a \rangle}$:

$$\pi^t_a(\theta_a) = \sum_{\mathbf{Z} \subseteq \mathbf{X} \setminus X_{\langle a \rangle}} \pi^t_{a|\mathbf{Z}}(\theta_a) P(\mathbf{Pa}_{\langle a \rangle} = \mathbf{Z} \mid \mathcal{D}[t]), \tag{3}$$

which is the key posterior distribution to be updated at each time step $t$ in the Bayesian CB problem. Note that if $Y \in \mathbf{Pa}^{\mathcal{G}}_{\langle a \rangle}$, then $P[Y = y \mid do(X_{\langle a \rangle} = x_a)] = P(Y = y)$ holds straightforwardly for $y \in \mathrm{Dom}(Y)$. Accordingly, if $Y \in \mathbf{Z}$, we compute $\hat{\mu}_{a,\mathrm{bda}}(\mathbf{Z})$ with the marginal distribution of $Y$ for the design of $\pi^0_{a|\mathbf{Z}}$.

**Structure Posterior**. The parent set distribution in (3) is obtained according to a posterior distribution of DAG structures informed by jointly observational and interventional data $\mathcal{D}[t]$:

$$P(\mathbf{Pa}_i = \mathbf{Z} \mid \mathcal{D}[t]) = \sum_{\mathcal{G}': \mathbf{Pa}^{\mathcal{G}'}_i = \mathbf{Z}} P(\mathcal{G}' \mid \mathcal{D}[t]). \tag{4}$$

The structure posterior is given by $P(\mathcal{G} \mid \mathcal{D}[t]) \propto P(\mathcal{D}[t] \mid \mathcal{G})P(\mathcal{G})$, where $P(\mathcal{G})$ is the structure prior, and the marginal likelihood $P(\mathcal{D}[t] \mid \mathcal{G}) = \int P(\mathcal{D}[t] \mid \mathcal{G}, \Theta_{\mathcal{G}})P(\Theta_{\mathcal{G}} \mid \mathcal{G})d\Theta_{\mathcal{G}}$ is obtained by integrating the likelihood function over the support of a conjugate prior of the parameters as follows. Let $m \in \mathcal{I} := \{1, \ldots, M\}$ index the $M = n_0 + t$ samples of data in $\mathcal{D}[t]$, and let $\mathcal{O}_i \subseteq \mathcal{I}$ represent the data points for which $X_i$ is not fixed by intervention. We make standard assumptions for Bayesian network structure learning, namely that the priors for the parameters of the conditional probability distributions satisfy global parameter independence, with $\Pi^0_{\mathcal{G}}(\Theta_{\mathcal{A}}) = \prod_{a \in \mathcal{A}} \pi^0_{a|\mathcal{G}}(\theta_a)$, as well as parameter modularity, with $\pi^0_{a|\mathcal{G}}(\theta_a) = \pi^0_{a|\mathcal{G}'}(\theta_a) = \pi^0_{a|\mathbf{Z}}(\theta_a)$ for graphs $\mathcal{G}$ and $\mathcal{G}'$ where $\mathbf{Pa}^{\mathcal{G}}_{\langle a \rangle} = \mathbf{Pa}^{\mathcal{G}'}_{\langle a \rangle} = \mathbf{Z}$ (see Heckerman et al. (1995) and Friedman & Koller (2003) for details). These allow us to express the marginal likelihood as $P(\mathcal{D}[t] \mid \mathcal{G}) = \prod_{i=1}^{p} P(x_i[\mathcal{O}_i] \mid \mathbf{pa}^{\mathcal{G}}_i[\mathcal{O}_i])$, where $x_i[\cdot]$ and $\mathbf{pa}^{\mathcal{G}}_i[\cdot]$ represent indexed samples of $X_i$ and $\mathbf{Pa}^{\mathcal{G}}_i$ in $\mathcal{D}[t]$, respectively. Assuming a conjugate prior, each conditional likelihood $P(x_i[\mathcal{O}_i] \mid \mathbf{pa}^{\mathcal{G}}_i[\mathcal{O}_i])$ can be calculated in closed form by integrating over the parameters:

$$P(x_i[\mathcal{O}_i] \mid \mathbf{pa}^{\mathcal{G}}_i[\mathcal{O}_i]) = \int \left[ \prod_{m \in \mathcal{O}_i} P\left(x_i[m] \mid \mathbf{pa}^{\mathcal{G}}_i[m], \theta^{\mathcal{G}}_i\right) \right] P(\theta^{\mathcal{G}}_i)d\theta^{\mathcal{G}}_i \tag{5}$$

where $\theta^{\mathcal{G}}_i = \theta_{X_i|\mathbf{Pa}^{\mathcal{G}}_i}$ is the parameters specifying the conditional distribution of $X_i$ given its parents (Eaton & Murphy, 2007).

Assuming the distribution $P$ is faithful to $\mathcal{G}$ (that is, all and only the conditional independence relationships in $P$ are entailed by $\mathcal{G}$), the posterior probability $P(\mathcal{G} \mid \mathcal{D}[t])$ will concentrate around the Markov equivalence class with increasing samples of observational data $n_0$. The equivalence class consists of the identification

of all direct edge connections (that is, the skeleton of $\mathcal{G}$) and some edge orientations called compelled edges, but even with infinite observational data, in general, not all edge orientations are identifiable without interventional data. The effect on $P(\mathcal{G} \mid \mathcal{D}[t])$ of pulling arm $a \in \mathcal{A}$ and observing interventional data according to the intervention $do(X_{\langle a \rangle} = x_a)$ is primarily though not limited to that of clarifying the orientation of the edges incident to $X_{\langle a \rangle}$ in $\mathcal{G}$.

Considering that the DAG space grows super-exponentially with the number of variables (Robinson, 1977), computation of the parent set probabilities $P(\mathbf{Pa}_i \mid \mathcal{D}[t])$ is admittedly challenging, even when the maximum number of parents is restricted. Due to the computational complexity, it is standard to assume a structure prior satisfying modularity, that is $P(\mathcal{G}) = \Pi_{i=1}^{p} P(\mathbf{Pa}_i^{\mathcal{G}})$, so that the posterior distribution is proportional to decomposable weights consisting of the product of local scores depending only on a node and its parents (Friedman & Koller, 2003). This property of score decomposability is crucial for the efficient implementation of Markov Chain Monte Carlo (MCMC) methods in which the probability distribution of features in $\mathcal{G}$ may be estimated by sampling DAGs from a Markov chain with stationary distribution $P(\mathcal{G} \mid \mathcal{D}[t])$ (Madigan et al., 1995; Friedman & Koller, 2003; Kuipers & Moffa, 2017; Kuipers et al., 2022). Particularly useful for our purposes is an algorithm developed by Pensar et al. (2020) to compute the exact parent set probabilities for a graph in time $O(3^p p)$ that also takes advantage of score decomposability. In our empirical evaluation of BBB, we apply BBB using both exact computation of parent set posteriors as well as approximation with MCMC sampling.

## 4 Bayesian Backdoor Bandit Algorithms

In this section, we apply our proposed BBB framework to several state-of-the-art MAB algorithms, namely upper confidence bound (UCB), Thompson sampling (TS), and Bayesian UCB (Bayes-UCB). Each method is concerned with designing and computing some criterion $U_a(t)$ to maintain a balance between exploration and exploitation when selecting arms according to $a_t \in \operatorname{argmax}_{a \in \mathcal{A}} U_a(t)$. In what follows, we briefly introduce these methods and discuss their application under the BBB framework. We then provide preliminary theoretical analysis of the cumulative regret for BBB-UCB and BBB-TS.

### 4.1 Description of Algorithms

The general UCB family of algorithms operates under the principle of optimism in the face of uncertainty (Lai & Robbins, 1985). Arms that have not been investigated as many times as others have more uncertain reward estimates and thus optimistically have potential for greater reward, motivating the design of a padding function $F_a(t)$ for computing the selection criterion $U_a(t) = \hat{\mu}_a(t) + F_a(t)$. Intuitively, the combination of the expected reward estimate $\hat{\mu}_a(t)$ and the uncertainty $F_a(t)$ maintains a balance between high confidence exploitation and potentially profitable exploration. Perhaps the most well-known and typically the default instantiation of UCB algorithms is UCB1 (Agrawal, 1995; Auer et al., 2002) which computes the following criterion:

$$U_a(t) = \hat{\mu}_a(t-1) + c\sqrt{\log(t-1)/n_a(t-1)}, \tag{6}$$

where $n_a(t) = \sum_{l=1}^{t} \mathbb{1}\{a_l = a\}$ denotes the number of times arm $a$ has been pulled in $t$ time steps. The confidence tuning parameter $c > 0$, discussed in Sutton & Barto (2018), controls the desired degree of exploration, where $c = \sqrt{2}$ in Auer et al. (2002). Hereafter, when discussing UCB, we refer to the policy expressed by the criterion in (6).

In what we refer to as BBB-UCB, we estimate the expected reward with the posterior mean $\mathrm{E}_{\pi_a^{t-1}}[\mu_a]$ with respect to (3), and we replace $1/n_a(t-1)$ with the posterior variance $\mathrm{Var}_{\pi_a^{t-1}}[\mu_a] \sim 1/(n_0 + n_a(t-1))$. In particular, for each arm $a \in \mathcal{A}$, we compute

$$U_a(t) = \mathrm{E}_{\pi_a^{t-1}}[\mu_a] + c\sqrt{\mathrm{Var}_{\pi_a^{t-1}}[\mu_a]\log(t)}, \tag{7}$$

where, for outer expectation with respect to the parent set distribution $\mathbb{P}_{t-1}(\mathbf{Z}) := P(\mathbf{Pa}_i = \mathbf{Z} \mid \mathcal{D}[t-1])$ in (4),

$$\mathrm{E}_{\pi_a^{t-1}}[\mu_a] = \mathrm{E}_{\mathbb{P}_{t-1}}\left[\mathrm{E}_{\pi_{a|\mathbf{Z}}^{t-1}}[\mu_a]\right], \quad \mathrm{Var}_{\pi_a^{t-1}}[\mu_a] = \mathrm{E}_{\mathbb{P}_{t-1}}\left[\mathrm{Var}_{\pi_{a|\mathbf{Z}}^{t-1}}[\mu_a]\right] + \mathrm{Var}_{\mathbb{P}_{t-1}}\left[\mathrm{E}_{\pi_{a|\mathbf{Z}}^{t-1}}[\mu_a]\right].$$

The Bayesian procedures of Bayes-UCB and TS are especially amenable to straightforward application under the BBB framework. These methods follow the Bayesian MAB formulation introduced in Section 2, typically taking as input uninformative priors in $\Pi^0$ that are equivalent for each arm. At each time step $t$, TS samples the expectations from the posterior $U_a(t) \leftarrow \pi_a^{t-1}(\mu_a)$, effectively selecting arm $a \in \mathcal{A}$ with probability equal to the posterior probability that $\mu_a$ is the highest expectation (Thompson, 1933). Bayes-UCB instead computes for each arm at time $t$ an upper quantile of $\mu_a$ based on its posterior distribution induced by $\pi_a^{t-1}$:

$$U_a(t) = Q\left(1 - \frac{1}{t(\log T)^c}, \pi_a^{t-1}\right), \tag{8}$$

where $Q(r, \rho)$ is the quantile function defining $P_\rho(X \leq Q(r, \rho)) = r$ for probability distribution $\rho$ and random variable $X \sim \rho$, and $c$ is a constant for computing the quantile used in the theoretical analysis of Bayes-UCB, with $c = 0$ empirically preferred (Kaufmann et al., 2012). For the BBB variants of Bayes-UCB and TS, we need simply to supply our designed backdoor adjustment prior $\Pi^0$ and make appropriate Bayesian updates to obtain the posterior $\Pi^t$.

We present our proposed BBB methodology applied to Bayes-UCB, TS, and UCB in Algorithm 1.

---

**Algorithm 1** `BBB-Alg`$(T, \mathcal{A}, \mathcal{D}_0, c)$

---

**Require:** Horizon $T$, action set $\mathcal{A}$, observational data $\mathcal{D}_0$, confidence level $c$
  1: Compute the observational parent set posteriors (4)
  2: **for all** $a \in \mathcal{A}$ and $\mathbf{Z} \subseteq \mathbf{X} \setminus X_{\langle a \rangle}$ **do**
  3:     Compute $\pi_{a|\mathbf{Z}}^0$ according to (2)
  4: **end for**
  5: **for all** $t = 1, \ldots, T$ **do**
  6:     **for all** $a \in \mathcal{A}$ **do**
  7:         Compute criterion $U_a(t)$ according to `Alg`:
             - `Bayes-UCB`: $U_a(t) = Q(1 - 1/(t(\log T)^c), \pi_a^{t-1})$ as in (8)
             - `TS`: Sample $U_a(t) \leftarrow \pi_a^{t-1}(\mu_a)$
             - `UCB`: $U_a(t) = \mathrm{E}_{\pi_a^{t-1}}[\mu_a] + c\sqrt{\mathrm{Var}_{\pi_a^{t-1}}[\mu_a]\log(t)}$ as in (7)
  8:     **end for**
  9:     Pull arm $a_t \in \mathrm{argmax}_{a \in \mathcal{A}} U_a(t)$ and observe $\mathcal{D}^{(t)}$
 10:     **for all** $\mathbf{Z} \subseteq \mathbf{X} \setminus X_{\langle a \rangle}$ where $a = a_t$ **do**
 11:         Update $\pi_{a|\mathbf{Z}}^t$ according to $\pi_{a|\mathbf{Z}}^t(\theta_a) \propto p_{\theta_a}(y_t)\, \pi_{a|\mathbf{Z}}^{t-1}(\theta_a)$
 12:     **end for**
 13:     Compute or update the parent set posteriors (4)
 14: **end for**

---

### 4.2 Preliminary Regret Analysis

In this section, we provide some preliminary analysis of the Bayesian cumulative regret for BBB-UCB and BBB-TS. We begin by defining the Bayesian cumulative regret. Given reward parameters $\Theta_{\mathcal{A}}$, the (expected) cumulative regret of a policy is defined as

$$R_T(\Theta_{\mathcal{A}}) = \mathrm{E}\left[\sum_{t=1}^{T}(\mu^* - \mu_{a_t}) \,\middle|\, \Theta_{\mathcal{A}}\right],$$

where the parameters $\Theta_{\mathcal{A}}$ are fixed. Under our Bayesian setting, we specify a prior $P(\mathcal{G})$ over the DAG $\mathcal{G}$ and conditional priors $\pi_{a|\mathbf{Z}}^0$ for the parameters of the reward distribution under interventions $a \in \mathcal{A}$, which defines a prior $\Pi^0$ over $\Theta_{\mathcal{A}}$. The Bayesian regret averages the regret $R_T(\Theta_{\mathcal{A}})$ over the prior distribution $\Pi^0$:

$$R_T^B = \mathrm{E}_{\Pi^0}\left[R_T(\Theta_{\mathcal{A}})\right] = \sum_{t=1}^{T} \mathrm{E}\left[\mu^* - \mu_{a_t}\right],$$

where the second expectation is with respect to the joint distribution over the parameters and the data $[\mathcal{G}, \Theta_{\mathcal{A}}, \mathcal{D}[T]]$. Note that $\mu_a = \mu_a(\Theta_{\mathcal{A}})$ is a random variable under the Bayesian setting. If $R_T^B = O(g(T))$, then $R_T(\Theta_{\mathcal{A}}) = O_p(g(T))$ with respect to the prior distribution of $(\mathcal{G}, \Theta_{\mathcal{A}})$.

Without loss of generality, assume $\mu_a \in [-C, C]$ for all $a \in \mathcal{A}$. We first analyze the BBB-UCB algorithm (Algorithm 1). By Eq. (2) in Russo & Van Roy (2014), for the UCB sequence $\{a_t\}$,

$$R_T^B \le \sum_{t=1}^{T} \mathrm{E}[U_{a_t}(t) - \mu_{a_t}] + 2C \sum_{t=1}^{T} P[\mu^* > U_{a^*}(t)]. \tag{9}$$

The first term $\mathrm{E}[U_{a_t}(t) - \mu_{a_t}] = \mathrm{E}\left[\mathrm{E}\left\{U_{a_t}(t) - \mu_{a_t} \mid \mathcal{D}[t-1]\right\}\right]$. Since $U_a(t)$ in (7) is a deterministic function of $\mathcal{D}[t-1]$, we have $\mathrm{E}\left\{U_{a_t}(t) \mid \mathcal{D}[t-1]\right\} = U_{a_t}(t)$ and consequently,

$$\begin{aligned} \mathrm{E}[U_{a_t}(t) - \mu_{a_t}] &= \mathrm{E}\left[U_{a_t}(t) - \mathrm{E}(\mu_{a_t} \mid \mathcal{D}[t-1])\right] \\ &= c\sqrt{\log t}\, \mathrm{E}\left[\sqrt{\mathrm{Var}(\mu_{a_t} \mid \mathcal{D}[t-1])}\right] \\ &\le c\sqrt{\log t}\, \sqrt{\mathrm{E}\left[\mathrm{Var}(\mu_{a_t} \mid \mathcal{D}[t-1])\right]}, \end{aligned} \tag{10}$$

where the last inequality follows from Jensen's inequality. Note that $\mathrm{Var}(\mu_a \mid \mathcal{D}[t]) = \mathrm{Var}_{\pi^t}(\mu_a)$ and $\mathrm{E}(\mu_a \mid \mathcal{D}[t]) = \mathrm{E}_{\pi^t}(\mu_a)$ as in (7).

We make the following assumptions on the posterior distribution $p(\mu_a \mid \mathcal{D}[t])$. See Appendix A for a detailed discussion on how to verify these assumptions.

**Assumption 1.** *Let $d_a$ be the number of candidate parent sets for $X_{\langle a \rangle}$. For all $t \ge 1$ and $a \in \mathcal{A}$,*

$$\mathrm{E}\left[\mathrm{Var}(\mu_a \mid \mathcal{D}[t])\right] \le \frac{c_1^2}{n_0 + n_a(t)} + c_2^2 d_a \exp(-\delta_a n_a(t)),$$

*where $c_1, c_2$ and $\delta_a$ are positive constants.*

The Markov equivalence class of the true DAG, represented by a CPDAG, can be accurately estimated with a large amount of observational data ($n_0$ is large). In such cases, $d_a \le 2^{m_a}$ is usually quite small, where $m_a$ is the number of undirected edges connected to $X_{\langle a \rangle}$ in the CPDAG.

The second assumption is on the concentration of $\mu^* \equiv \mu_{a^*}$ around its posterior mean $\mathrm{E}(\mu_{a^*} \mid \mathcal{D}[t])$:

**Assumption 2.** *For all $t \ge 1$,*

$$P\left\{\frac{\mu^* - \mathrm{E}(\mu^* \mid \mathcal{D}[t])}{\sqrt{\mathrm{Var}(\mu^* \mid \mathcal{D}[t])}} > c\sqrt{\log t} \,\Big|\, \mathcal{D}[t]\right\} \le c_3 t^{-b},$$

*where $c_3 > 0$ and $b > 1$ are constants.*

**Proposition 1.** *Under Assumptions 1 and 2, the Bayes regret of the BBB-UCB algorithm satisfies*

$$R_T^B \le \left[c_4\left(\sqrt{KT + K^2(n_0 - 1)} - \sqrt{K^2(n_0 - 1)}\right) + c_5 \sum_{a \in \mathcal{A}} \sqrt{d_a}\right]\sqrt{\log T} + c_6, \tag{11}$$

*where $K = |\mathcal{A}|$ and $c_4, c_5, c_6$ are positive constants.*

**Remark 1.** *By Proposition 1 of Russo & Van Roy (2014), the same upper bound* (11) *also applies to the Bayes regret of the BBB-TS algorithm.*

For any $n_0 \geq 1$, by concavity of $\sqrt{x}$,

$$\sqrt{K^2(n_0 - 1) + KT} - \sqrt{K^2(n_0 - 1)} \leq \sqrt{KT}.$$

Therefore, when $T$ is large such that $\sum_a \sqrt{d_a} = O(\sqrt{T})$, the regret $R_T^B = O(\sqrt{KT \log T})$, which is identical to the order of the regret of a standard MAB, e.g. Proposition 2 of Russo & Van Roy (2014). The benefit of our backdoor adjustment prior is seen when $n_0$ is large relative to $T$. If $T/(n_0 - 1) < M$, where $M$ is a constant, then

$$\sqrt{K^2(n_0 - 1) + KT} - \sqrt{K^2(n_0 - 1)} = \sqrt{K^2(n_0 - 1)} \left[ \left\{ 1 + \frac{T}{K(n_0 - 1)} \right\}^{1/2} - 1 \right]$$

$$\leq \frac{T}{2\sqrt{n_0 - 1}} \leq \frac{\sqrt{MT}}{2},$$

where the first inequality is due to $(1 + x)^{1/2} \leq 1 + x/2$ for $x \geq 0$. In this case, we obtain a regret bound $R_T^B = O(\sqrt{T \log T})$ independent of $K$. This confirms the advantage of using observational data to *simultaneously* estimate all rewards $\mu_a, a \in \mathcal{A}$ through backdoor adjustment, which largely relaxes the dependence of the regret on the number of actions.

## 5 Implementation Details

### 5.1 Nonparametric Discrete Setting

We now detail the application of our proposed construction of $\pi_{a|\mathbf{Z}}^0$ to the setting where the conditional probability distributions are multinomials, with each variable $X_i \in \mathbf{X}$ probabilistically attaining its states depending on the attained state configuration of its parents $\mathbf{Pa}_i^{\mathcal{G}}$. The reward variable $Y = X_p$ is a binary variable with $\mathrm{Dom}(Y) = \{0, 1\}$. If $Y \notin \mathbf{Z}$, $\mu_a$ may be estimated with observational data through straightforward empirical estimation of (1):

$$\hat{\mu}_{a,\mathrm{bda}}(\mathbf{Z}) = \hat{P}[Y = 1 \mid do(X_{\langle a \rangle} = x_a)] = \frac{1}{n_0} \sum_{\mathbf{z}} \frac{n_0[1, x_a, \mathbf{z}] n_0[\mathbf{z}]}{n_0[x_a, \mathbf{z}]}, \tag{12}$$

where $n_0[1, x_a, \mathbf{z}]$ represents the number of the $n_0$ samples of $\mathcal{D}_0$ in which $Y = 1$, $X = x_a$, and $\mathbf{Z} = \mathbf{z}$, with corresponding definitions for $n[x_a, \mathbf{z}]$ and $n[\mathbf{z}]$.

Analysis of the sampling distribution of (12) is admittedly challenging. To design an appropriately weighted informative prior as proposed in (2), we require some characterization of the sampling variability of $\hat{\mu}_{a,\mathrm{bda}}(\mathbf{Z})$. Hence, we derive an approximation of the variance of (12), $\widehat{\mathrm{SE}}^2[\hat{\mu}_{a,\mathrm{bda}}(\mathbf{Z})]$. We accomplish this by first re-expressing the joint counts $n_0[\cdot]$ as sums of elements of a multinomial random vector. The term within the sum may then be expressed as a product and ratio of intersecting random quantities, which we approximate through a first-order Taylor series expansion. The details of the derivation are delegated to Appendix D. It is appropriate to acknowledge that Maiti et al. (2021) proposed a provably unbiased strategy for empirical estimation of (1) through splitting the sample into independent partitions. However, this approach suffers from severe loss of precision through what some may consider underutilization of the observed data. In our experiments detailed in Appendix C.1, we find the empirical performance of (12) in our applications to be acceptable. We additionally provide extensive empirical validation of our derived approximation, demonstrating coverage probabilities comparable to empirical estimates of the sampling variability for modest sample sizes.

Since the reward variable under arm $a$ is a Bernoulli random variable with probability parameter $\mu_a = P[Y = 1 \mid do(X_{\langle a \rangle} = x_a)]$, we assume a conjugate prior $\pi_{a|\mathbf{Z}}^0 = \mathrm{Beta}(\alpha_0, \beta_0)$ for $\theta_a = \mu_a$ designed according

to (2), resulting in prior hyperparameters

$$\alpha_0 = \hat{\mu}_{a,\text{bda}}(\mathbf{Z}) \left( \frac{\hat{\mu}_{a,\text{bda}}(\mathbf{Z})[1 - \hat{\mu}_{a,\text{bda}}(\mathbf{Z})]}{\hat{\text{SE}}^2[\hat{\mu}_{a,\text{bda}}(\mathbf{Z})]} - 1 \right), \qquad \beta_0 = \alpha_0 \left( \frac{1 - \hat{\mu}_{a,\text{bda}}(\mathbf{Z})}{\hat{\mu}_{a,\text{bda}}(\mathbf{Z})} \right).$$

## 5.2 Gaussian Unit Deviation Setting

In this section, we consider the setting where the causal model may be expressed as a set of Gaussian structural equations:

$$X_j = \sum_{i=1}^p \beta_{ij} X_i + \varepsilon_j, \quad \varepsilon_j \sim \text{N}(0, \sigma_j^2), \quad j = 1, \ldots, p. \tag{13}$$

There is no intercept term, which is analogous to having prior knowledge of the observational means, and we consider interventions $x_a \in \{-1, 1\}$, which may be interpreted as investigating unit deviations from the observational means. In this setting, the causal effect of $X_{\langle a \rangle}$ on $Y$ is given by

$$\psi_{\langle a \rangle} := \text{E}_P[Y \mid do(X_{\langle a \rangle} = x' + 1)] - \text{E}_P[Y \mid do(X_{\langle a \rangle} = x')]$$

for any $x' \in \mathbb{R}$, derived via a special case of (1). Note that in our problem formulation, $Y \mid do(X_{\langle a \rangle} = 1)$ and $-Y \mid do(X_{\langle a \rangle} = -1)$ are identically distributed, so all data generated from interventions on $X_{\langle a \rangle}$ may be combined to estimate $\psi_{\langle a \rangle}$. Since $\mu_a = x_a \psi_{\langle a \rangle}$, we focus our efforts on estimating and modeling $\psi_{\langle a \rangle}$. Accordingly, in constructing our priors using intervention calculus, we design priors $\pi^0_{\langle a \rangle | \mathbf{Z}}$ for $\theta_{\langle a \rangle}$ corresponding to estimating $\psi_{\langle a \rangle}$, and allow $\pi^0_{a | \mathbf{Z}}$ to be the induced priors for $\theta_a$ corresponding to $\mu_a = \psi_{\langle a \rangle} x_a$, detailed as follows.

If $Y \notin \mathbf{Z}$, then a consistent estimator of $\psi_{\langle a \rangle}$, denoted $\hat{\psi}_{\langle a \rangle, \text{bda}}$, may be obtained with observational data by the least squares regression

$$Y = \psi_{\langle a \rangle} X_{\langle a \rangle} + \boldsymbol{\gamma}^\top \mathbf{Z} + e, \quad e \sim \text{N}(0, \eta^2), \tag{14}$$

where $\boldsymbol{\gamma} \in \mathbb{R}^{|\mathbf{Z}|}$ is the coefficients of the parents $\mathbf{Z}$ (Maathuis et al., 2009; Pensar et al., 2020), and some dependence on $a$ is omitted for simplicity. Correspondingly, we express the desired interventional distribution as $Y \mid do(X_{\langle a \rangle} = x_a) \sim \text{N}(\psi_{\langle a \rangle} x_a, \omega^2)$. Claiming no prior knowledge of the interventional variance, we assume a Normal-inverse-gamma (N-$\Gamma^{-1}$) conjugate prior $\pi^0_{\langle a \rangle | \mathbf{Z}}$ for $\theta_{\langle a \rangle} = (\psi_{\langle a \rangle}, \omega^2)$:

$$\psi_{\langle a \rangle} \mid \omega^2 \sim \text{N}\left(m_0, \omega^2 \nu_0^{-1}\right), \quad \omega^2 \sim \Gamma^{-1}(u_0, v_0). \tag{15}$$

Since in general, the residual variance $\eta^2$ in (14) is not equivalent to $\omega^2$, we propose the following to estimate $\omega^2$ from observational data.

**Proposition 2.** *Suppose that* $\mathbf{X}$ *follows the causal structural equation model (SEM) in* (13) *with CBN* $\mathcal{B}$. *Let* $Y, X \in \mathbf{X}$, *and denote by* $\psi$ *the causal effect of* $X$ *on* $Y$. *Then for any* $x \in \text{Dom}(X)$,

$$\text{Var}_P[Y \mid do(X = x)] = \text{Var}_P[Y - \psi X].$$

Note that $\text{Var}_P[\cdot]$ is the variance in $P$ defined by CBN $\mathcal{B}$, and the variance on the right side is with respect to the observational distribution of $\mathbf{X}$. Intuitively, subtracting by $\psi X$ negates the noise variances $\sigma^2$ in (13) propogated through and from $X$ to $Y$. We include a detailed proof for Proposition 2 in Appendix A.

Thus, to estimate $\omega^2$ from $\mathcal{D}_0$, we propose the estimator $\hat{\omega}^2 = \sum_i (\tilde{y}_i - \bar{\tilde{y}})^2 / (n_0 - |\mathbf{Z}| - 2)$ where $\tilde{y}_i$ are realizations of $\tilde{Y} := Y - \hat{\psi}_{\langle a \rangle, \text{bda}}(\mathbf{Z}) X_{\langle a \rangle}$ in $\mathcal{D}_0$, and $n_0 - |\mathbf{Z}| - 2$ is the degrees of freedom resulting from estimating $\bar{\tilde{y}}$ in addition to $|\mathbf{Z}| + 1$ coefficients in (14). Accordingly, we design the prior $\omega^2 \sim \Gamma^{-1}(u_0, v_0)$ to have prior mean $\text{E}[\omega^2] = v_0 / (u_0 - 1) = \hat{\omega}^2$, resulting in hyperparameters $u_0 = (n_0 - |\mathbf{Z}|)/2$ and $v_0 = \sum_i (\tilde{y}_i - \bar{\tilde{y}})^2 / 2$. After marginalizing out $\omega^2$, $\psi_{\langle a \rangle} \sim t_{2u_0}(m_0, v_0(u_0 \nu_0)^{-1})$, so we set $\text{E}_{\pi^0_{\langle a \rangle | \mathbf{Z}}}[\psi_{\langle a \rangle}] = m_0 = \hat{\psi}_{\langle a \rangle, \text{bda}}(\mathbf{Z})$ and solve to obtain $\nu_0 = v_0 / (u_0 \hat{\text{SE}}^2[\hat{\psi}_{\langle a \rangle, \text{bda}}(\mathbf{Z})])$.

To maximally utilize the ensemble data, we further generalize the estimation of $\psi_{\langle a \rangle}$ via regression in (14) to include eligible samples of intervention data. This is achieved through the following proposition, which we prove in Appendix A. This result does not rely on any parametric assumptions for the underlying causal model, assuming simply that $\mathbf{X}$ follows a general linear SEM with DAG $\mathcal{G}$ (Pearl, 2000).

**Proposition 3.** *Suppose that* $\mathbf{X}$ *follows a linear SEM with CBN* $\mathcal{B} = (\mathcal{G}, \Theta_{\mathcal{G}})$, *and* $X, Y \in \mathbf{X}$. *Suppose that* $W \in \mathbf{X} \setminus \{X, Y\}$ *does not block any directed path from* $X$ *to* $Y$ *in* $\mathcal{G}$. *Then for any* $w \in \mathbb{R}$,

$$\frac{\partial}{\partial x} \mathrm{E}_P[Y \mid do(X = x)] = \frac{\partial}{\partial x} \mathrm{E}_P[Y \mid do(X = x), do(W = w)].$$

Proposition 3 asserts a simple graphical criterion which, if satisfied, defines an avenue by which information can be shared between arms. In our work, we check the graphical criterion for estimating the causal effect of $X_{\langle a \rangle}$ on $Y$ with interventional data generated from intervening on $X_j$ as follows. Using another algorithm proposed by Pensar et al. (2020) for computing exact ancestor posterior probabilities, we consider the criterion satisfied at time step $t$ if the event that $X_j$ blocks a directed path from $X_{\langle a \rangle}$ to $Y$ has low posterior probability:

$$P(X_{\langle a \rangle} \rightsquigarrow X_j \rightsquigarrow Y \mid \mathcal{D}[t]) \leq \min\{P(X_{\langle a \rangle} \rightsquigarrow X_j \mid \mathcal{D}[t]), P(X_j \rightsquigarrow Y \mid \mathcal{D}[t])\} \leq \tau \tag{16}$$

where $X_j \rightsquigarrow Y$ denotes that $X_j$ is an ancestor of $Y$ and the threshold is set to $\tau = 0.1$ in our application. If (16) holds at time step $t$, we combine the observational data and the data from interventions on $X_j$ when conducting the regression (14). While independent samples of observational and interventional data are not guaranteed to have identically distributed errors in the regression (14), we provide extensive empirical validation of our proposed regression with ensemble data in Appendix C.2, confirming indistinguishable performance for the purposes of estimating $\psi_{\langle a \rangle}$ and its sampling variability compared to that of purely observational data.

## 6  Numerical Experiments

We conducted extensive numerical experiments to empirically validate our proposed methodology. For our main experiments, we generated random CBN models in an effort to empirically demonstrate the merits of our proposed methodology by evaluating BBB across a broad range of scenarios. To address the computational challenges of BBB, we then evaluated an implementation using MCMC to estimate parent set probabilities and applied it to a larger realistic reference network. Comprehensive experimental details sufficient for reproducing our experiments, such as CBN preparation and algorithm parameters, are provided in Appendix B. Additional supplemental experiments independently evaluating (12) and Proposition 3 may be found in Appendix C. The complete code and instructions for reproducing our results have been made available at the following link:

```
https://github.com/jirehhuang/bcb
```

### 6.1  Random Networks

For our main experiments, we generated CBN models with $p = 10$ variables in the interest of representing a diverse range of scenarios in our empirical comparisons. The structures were randomly generated, with the reward variable designated to have $|\mathbf{Pa}_p^{\mathcal{G}}| = 3$ parents, according to a process adapted from de Kroon et al. (2022). The conditional probability distributions of each CBN were likewise generated randomly. Atomic interventions as described in Section 5 were allowed on all variables excluding the reward variable, with the discrete variables assumed to be binary, resulting in $|\mathcal{A}| = 2(p - 1) = 18$ actions.

We evaluated our BBB methodology against algorithms designed to optimize cumulative regret, including popular standard MAB algorithms Bayes-UCB, TS, and UCB that do not utilize causal assumptions (see Section 4). Additionally, we compared against what can be interpreted as a highly optimistic version of the central node approach by Lu et al. (2021), introduced in Section 1, by presupposing knowledge of the direct causes of the reward variable. In particular, for Bayes-UCB*, TS*, and UCB*, we executed the

respective algorithms over the reduced action set $\mathcal{A}' = \{a \in \mathcal{A} : \langle a \rangle \in \mathbf{Pa}_p^{\mathcal{G}}\}$. Accordingly, for the cases where $\langle a^* \rangle \notin \mathbf{Pa}_p^{\mathcal{G}}$, we redefined the optimal intervention to $a^* = \mathrm{argmax}_{a \in \mathcal{A}'} \mu_a$ when evaluating the regret of TS* and (Bayes-)UCB*. To circumvent confounding arising from the differences in algorithm designs and any relevant parameter tuning, we focus our comparisons within algorithm types, e.g. amongst TS, TS*, and BBB-TS.

Using the process described above, we generated 100 CBN models for each distributional setting. For each CBN, we executed the competing methods 10 times but our BBB methods only 5 times due to their greater computational expense, with $T = 5000$ time steps. The results presented are averaged across all simulations for each time step, with the cumulative regret normalized by the optimal reward $\mu^*$ to ensure that each CBN model contributes comparably. In preference to the competing methods, we tuned for their best-performing parameters where relevant and applied them to our BBB implementations. We computed exact parent set probabilities in BBB using an algorithm proposed by Pensar et al. (2020) in an effort to assess our BBB methodology in the most precise implementation of its formulation. Additional experimental details may be found in Appendix B, and additional supporting results are provided in Appendix C.

Note that while the simulations were executed across a cluster of heterogeneous nodes, it was not difficult to observe that the computational requirements of BBB vastly exceeded that of the considered competing methods. The average execution time of each iteration of our discrete and Gaussian implementations of BBB on the random networks was respectively around 4.3 and 8.2 seconds, compared to less than 0.15 seconds for the competing methods. This includes the exact computation of parent set posteriors, computing or updating corresponding backdoor adjustment estimates, and in the Gaussian case, ancestor posterior probabilities. While BBB requires significantly greater computational expense, the cost must be weighed against other considerations such as the time and resources required for additional exploration of interventions.

### 6.1.1 Cumulative Regret Comparisons

The empirical cumulative regret results in Figure 1 demonstrate that in both the discrete and Gaussian settings and for all algorithms, our BBB methodology is able to reliably outperform the non-causal variants with finite samples of observational data. The improvement increases monotonically with increasing sample sizes of observational data $(n_0)$. While corresponding variants of Bayes-UCB and TS perform comparably, UCB achieves substantially lower regret because the parameter $c$ in (6) was tuned to maintain a balance between exploration and exploitation that is most empirically preferred. In particular, UCB is able to avoid excessive exploration by scaling its padding term with a relatively small constant, whereas Bayes-UCB maintains a relatively high minimum exploration rate according to its formulation in (8), as does TS.

In comparison to the optimistic central node versions of the algorithms, BBB generally achieves lower cumulative regret with $n_0 \geq 800$ in the discrete setting and $n_0 \geq 40$ in the Gaussian setting. Recall that, in practical applications, the central node approach relies on the availability of large-sample observational data as well as a sequence of interventions to recover the reward generating variables $\mathbf{Pa}_p^{\mathcal{G}}$. Based on our simulation settings, this reduces the action set from $|\mathcal{A}| = 18$ arms to only $|\mathcal{A}'| = 2|\mathbf{Pa}_p^{\mathcal{G}}| = 6$ arms, and we additionally artificially restrict $a^* \in \mathcal{A}'$ to evaluate the regret. Furthermore, the regret results reported for these methods do not include the interventions required to identify $\mathbf{Pa}_p^{\mathcal{G}}$, thus representing a kind of best case scenario for the central node approach. In contrast, our methodology derives substantial benefit from modest amounts of observational data samples $n_0$.

Indeed, we find that our BBB methods are able to perform competitively against the competing methods even when the latter are given $n_0$ time steps to explore arms before incurring regret. To compensate for the fact that BBB utilizes $n_0$ samples of observational data prior to investigating arms, we present the results where the competing algorithms TS(*) and (Bayes-)UCB(*) are given a head start of $n_0 \in \{100 \cdot 2^k : k = 0, 1, \ldots, 5\}$ time steps to explore arms before incurring regret. The results for the discrete setting are shown in Figure 2. The Gaussian results are omitted because $n_0 \leq 320$ is relatively small, so the head start does not offer substantial benefit to the competing methods. In all cases, BBB still significantly outperforms the standard algorithms TS and (Bayes-)UCB given the head start. Given sufficient samples of observational data, BBB still performs comparably to if not better than the optimistic central node variants in terms of cumulative

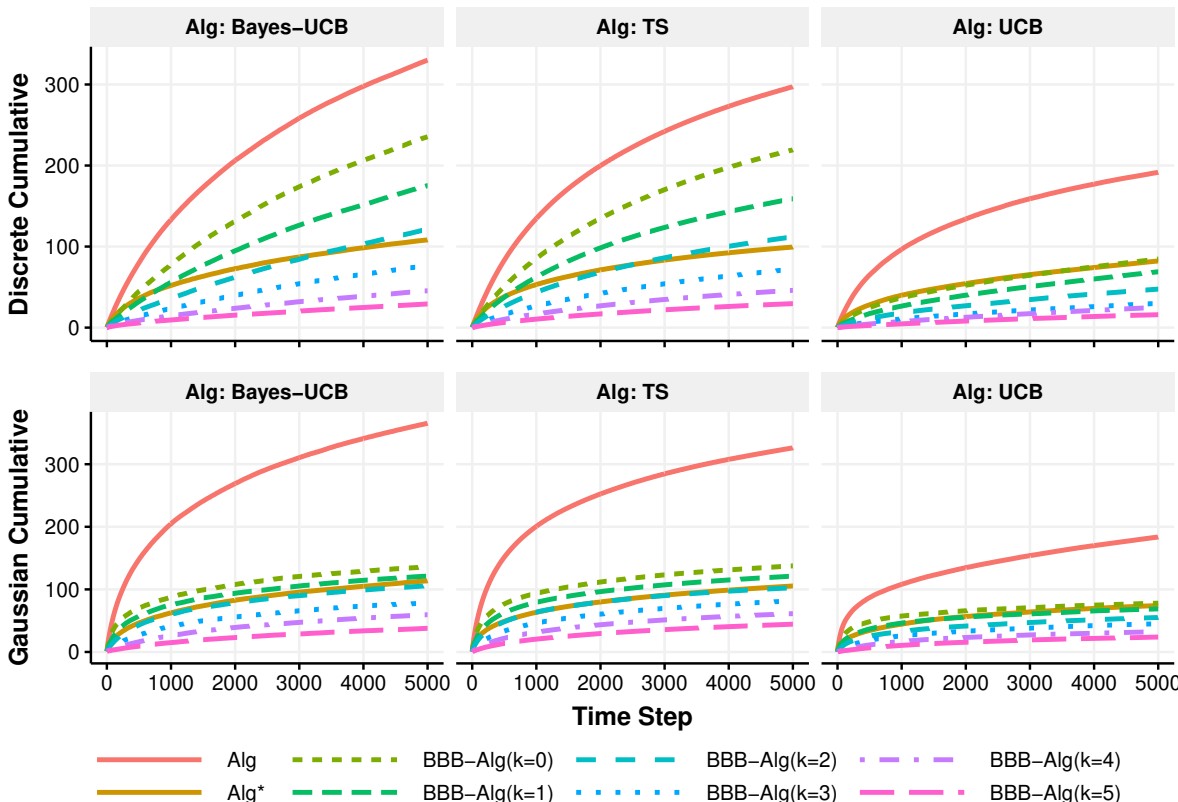

Figure 1: Average cumulative regret against $T = 5000$ time steps comparing Alg, Alg$^*$, and BBB-Alg for Alg $\in \{\text{Bayes-UCB}, \text{TS}, \text{UCB}\}$. BBB methods were executed with $n_0 = 100 \cdot 2^k$ in the discrete setting and $n_0 = 10 \cdot 2^k$ in the Gaussian setting.

regret, for which the head start is only an additional unwarranted advantage given that they already require significantly more observational data as well as additional interventions.

### 6.1.2 Structure Identification

In addition to the cumulative regret performance, it is of interest to consider the structure identification behavior of the BBB approach in our experiments. We measure the concentration of the posterior probability across DAGs $\mathcal{G}$ with respect to the underlying causal graph $\mathcal{G}^*$ using the edge support sum of absolute errors (ESSAE), which is given at time $t$ by

$$\sum_{i=1}^{p} \sum_{j \neq i} \left| P(j \in \mathbf{Pa}_i^{\mathcal{G}} \mid \mathcal{D}[t]) - \mathbb{1}\left\{ j \in \mathbf{Pa}_i^{\mathcal{G}^*} \right\} \right|.$$

This quantity may be understood as a probabilistic version of the structural hamming distance, a common metric in Bayesian network structure learning literature. Lower ESSAE corresponds to greater concentration of the posterior probability around the causal graph $\mathcal{G}^*$. The results are provided in Figure 3.

In the discrete results for BBB-Bayes-UCB and BBB-TS, the initial ESSAE is unsurprisingly lower for the larger sample sizes, but the trend quickly reverses as the time steps progress. This effect is also observed occurring in the Gaussian results, but at an accelerated pace. This behavior is perhaps best understood in complement to the cumulative regret results in Figure 1. If $P$ is faithful to $\mathcal{G}$, then if $n_0$ is large, the structure prior $P(\mathcal{G} \mid \mathcal{D}_0)$ is expected to concentrate around the Markov equivalence class, which entails identification of the skeleton and in general, partial identification of the orientations. Additionally, the conditional priors $\pi_{a|\mathbf{z}}^0$ will be precise models, allowing BBB to quickly identify and select arm(s) $a \in \mathcal{A}$ with small regret

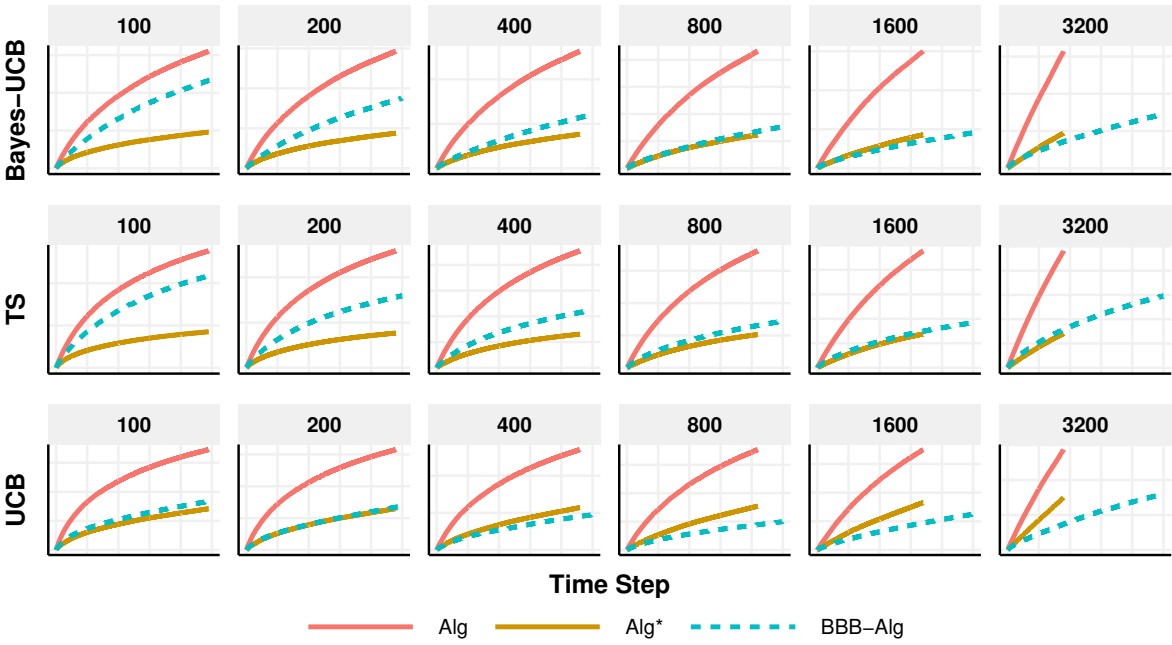

Figure 2: Discrete average cumulative regret for $\text{Alg} \in \{\text{Bayes-UCB}, \text{TS}, \text{UCB}\}$ with a head start of $n_0 \in \{100 \cdot 2^k : k = 0, 1, \ldots, 5\}$ time steps for competing methods.

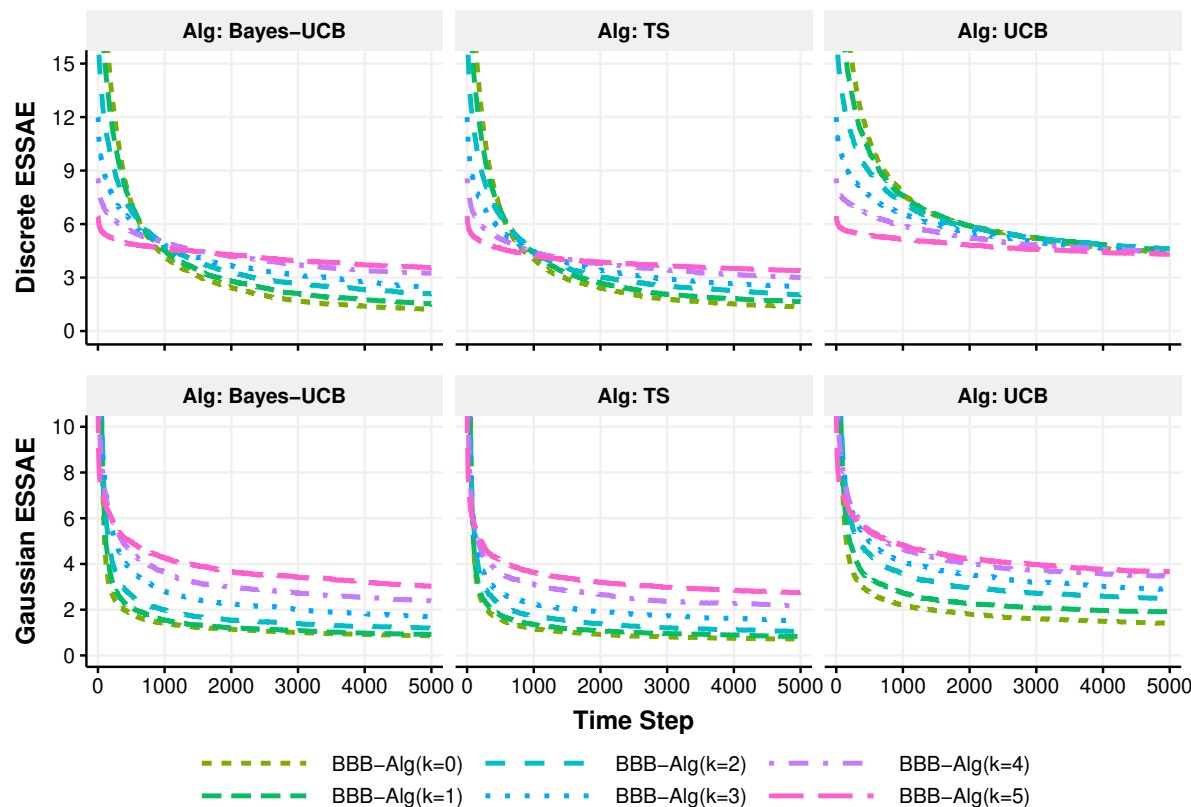

Figure 3: Discrete ($n_0 = 100 \cdot 2^k$) and Gaussian ($n_0 = 10 \cdot 2^k$) results of the average ESSAE of the full graph structure for BBB-Alg, $\text{Alg} \in \{\text{Bayes-UCB}, \text{TS}, \text{UCB}\}$.

$\mu^* - \mu_a$, which has the effect of clarifying the orientation of edges incident to such $X_{\langle a \rangle}$. The policies take no interest in determining the orientation of the remaining edges if the uncertainty does not indicate potential to identify more profitable actions. In contrast, when $n_0$ is small, the greater uncertainty in both the structure prior and the conditional priors encourage the exploration of many different arms, thus incurring greater cumulative regret. In addition to clarifying the orientation of the incident edges, selecting arm(s) $a \in \mathcal{A}$ contributes to identifying the direct edge connections excluding those from $\mathbf{Pa}_{\langle a \rangle}^{\mathcal{G}}$ to $X_{\langle a \rangle}$, as can be seen in (5). Thus, the skeleton is recovered and more edge orientations are identified than in the case where $n_0$ is large, achieving lower ESSAE at the cost of greater cumulative regret. Notably, while this reversal appears to be absent in the discrete results for BBB-UCB, in actuality it has simply not yet been realized even after $T = 5000$ time steps due to the small exploration constant $c$ in (7).

## 6.2 Scaling BBB With MCMC

Despite the efficiency of the algorithm proposed by Pensar et al. (2020), its scaling in $p$ of $O(3^p p)$ means that exact computation of the parent set posteriors is not always feasible. Pensar et al. (2020) noted that their algorithm executed on 20 variables in about 25 minutes, which would translate to over 86 days for 5000 time steps. While potentially justifiable depending on the context of the application such as when interventions are particularly expensive or time-consuming, such intensive computational requirement for each time step is likely to be limiting of the practical application of BBB on larger systems. As discussed in Section 3, the parent set probabilities are derived from a posterior distribution of graph structures which may be estimated by MCMC. In this section, we discuss the implementation of BBB using MCMC to estimate the structure posterior. We provide preliminary results in the discrete setting assessing this approximation against the exact computation of parent sets and evaluating its performance on a realistic reference network with $p = 20$ variables.

The posterior distribution of graph structures $P(\mathcal{G} \mid \mathcal{D}[t])$ may be approximated by sampling DAGs $\mathbb{G}^t$ from $P(\mathcal{G} \mid \mathcal{D}[t])$ using MCMC and empirically estimating (4) from the sampled graphs:

$$P(\mathbf{Pa}_i = \mathbf{Z} \mid \mathcal{D}[t]) \approx \frac{1}{|\mathbb{G}^t|} \sum_{\mathcal{G}' \in \mathbb{G}^t} \mathbb{1}\left\{ \mathbf{Pa}_i^{\mathcal{G}'} = \mathbf{Z} \right\}.$$

Various MCMC sampling schemes for DAGs have been developed, most basic of which is the structure MCMC sampler which accepts proposed single edge addition, deletion, or reversal steps according to a Metropolis-Hastings probability (Madigan et al., 1995; Giudici & Castelo, 2003). With order MCMC, Friedman & Koller (2003) reduced the search space to the topological orderings of the $p$ nodes, significantly accelerating convergence but retaining a degree of bias since a given DAG may belong to multiple orders. Kuipers & Moffa (2017) addressed this issue of bias by considering the space of ordered partitions in partition MCMC, though at the price of greater computational requirement (Suter et al., 2021).

In our simulations, we chose to use order MCMC despite the potential bias due to its computational advantage over partition MCMC while achieving satisfactory performance. For further improvements in efficiency, we applied the hybrid approach proposed by Kuipers et al. (2022) in which DAGs are sampled from a loosely restricted initial search space, with built-in provisions to expand the search space. In particular, before performing any interventions, we obtained an initial search space with the PC algorithm (Spirtes & Glymour, 1991) and sampled DAGs from $P(\mathcal{G} \mid \mathcal{D}_0)$. For subsequent iterations, we restricted the search space using the structure posterior estimated from the immediately preceding iteration. Whenever conducting the restricted sampling scheme, the search space was allowed to be extended as designed by Kuipers et al. (2022) to account for false negatives in the restriction.

We first compared the cumulative regret amongst exact computation of parent set posteriors (Exact), approximation by MCMC without restricting the search space at any step (MCMC), and MCMC with restricting the search space as described (Hybrid MCMC). Each of these were executed twice on each of the 100 randomly generated discrete CBNs, and the results at various time steps are shown in Figure 4. Hybrid MCMC performed equivalently with unrestricted MCMC, enjoying significant computational reductions without exhibiting any inferiority in performance. However, while (hybrid) MCMC performed comparably with exact computation in most cases, there were significantly more extreme values in the MCMC methods. These

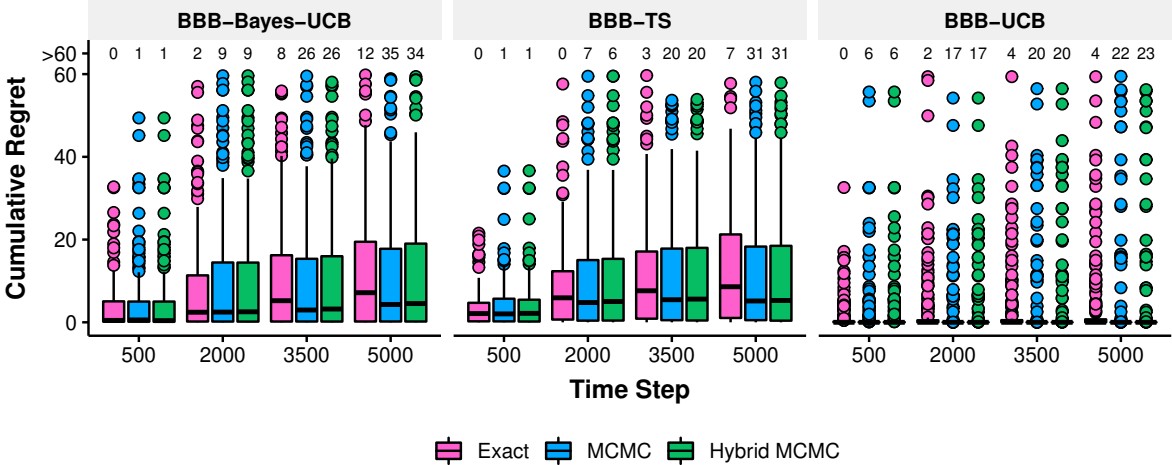

Figure 4: Cumulative regret of BBB-Alg for Alg $\in$ {Bayes-UCB, TS, UCB} with $n_0 = 3200$ on the discrete random networks from Section 6.1 for time steps $t \in \{500, 2000, 3500, 5000\}$ comparing between computing exact parent set posteriors and approximating with MCMC. Each boxplot represents 200 data points, with the number of points above 60 labeled above.

may have resulted from the bias inherent to order MCMC, but given that general settings were applied and convergence was not assessed for each iteration, another potential explanation would be insufficient iterations for certain CBNs. In general, approximation of the structure posterior with MCMC appears to be an acceptable scalable alternative to exact computation.

We then applied BBB with hybrid MCMC to CHILD, a moderately sized discrete reference network with $p = 20$ nodes for diagnosing congenital heart disease (Spiegelhalter, 1992). CHILD was constructed with domain experts and made available in a Bayesian network repository (Spiegelhalter, 1992; Scutari, 2010). The network was preprocessed to have binary variables, and the target variable was set to LowerBodyO2, a leaf node with two parents. All other variables were intervenable, with $|\mathcal{A}| = 38$ arms, though only $|\mathcal{A}'| = 4$ for the optimistic central node approach.

The cumulative regret results for 20 executions of various algorithms are presented in Figure 5. Even after $T = 5000$ time steps, most methods fail to exhibit meaningful convergence in the form of the flattening of the cumulative regret curve. This is because all but two arms have expected reward $\mu_a > 0.5$, making the optimal reward of approximately $\mu^* = 0.622$ difficult to distinguish without a great deal of exploration. Unsurprisingly, the central node approach performs exceptionally well, having only four arms to investigate. With $n_0 = 3200$, BBB was able to adequately distinguish the arms with higher expected reward, achieving cumulative regret competitive with the central node approach. The average execution time of each iteration of BBB with MCMC on CHILD was around 9.4 seconds, compared to less than 0.23 seconds for the competing methods.

# 7 Discussion

In this paper, we proposed the BBB framework for enhancing experimental investigations with observational data. BBB consists of an aggregation of various strategies for estimating and modeling the parameters of interest with jointly interventional and observational data in order to efficiently utilize all available data to inform exploitation and exploration. Applied in our methodology but also of independent interest, we derived a well-performing approximation for the variance of the discrete backdoor adjustment estimator, and in the Gaussian setting, we characterized the interventional variance using the observational distribution and proposed a simple graphical criterion for sharing information between arms. We supplied preliminary regret analysis justifying our methodology, and empirically validated our proposed algorithms through extensive

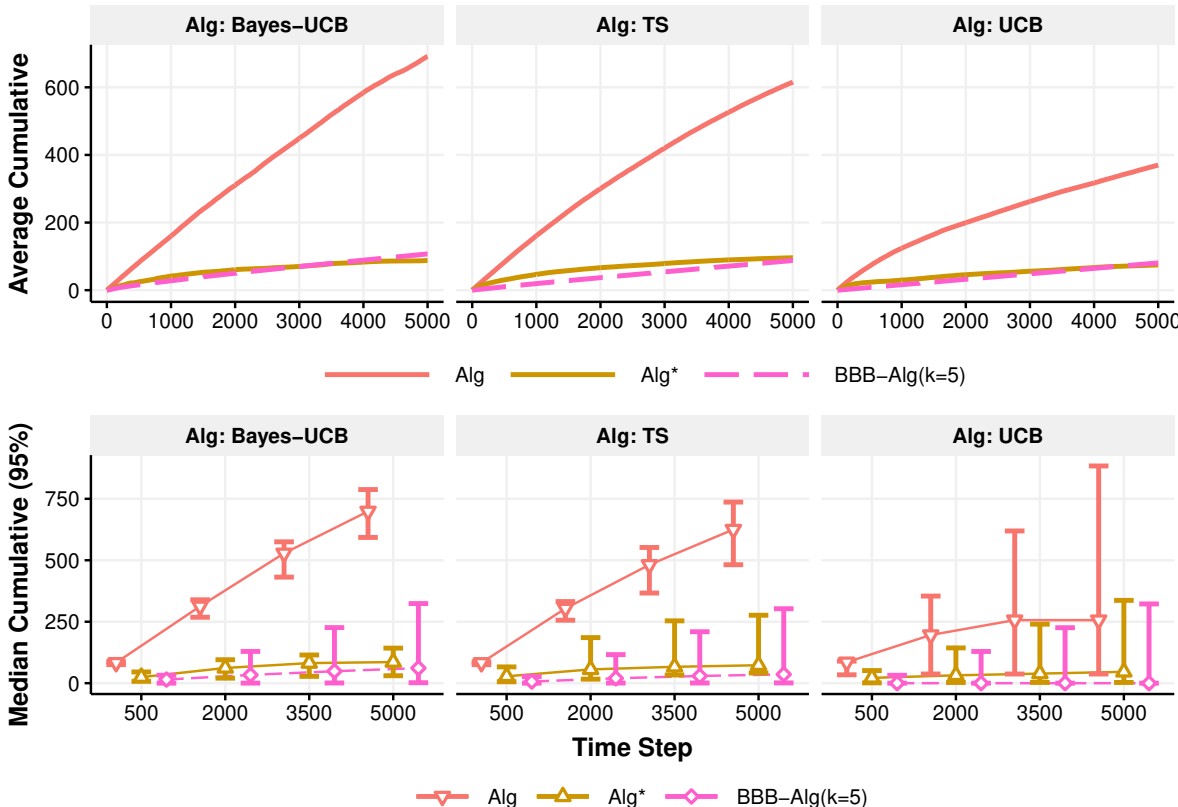

Figure 5: Average cumulative regret and median cumulative regret with 95% percentile intervals against $T = 5000$ time steps comparing Alg, Alg*, and BBB-Alg for Alg $\in \{$Bayes-UCB, TS, UCB$\}$, executed on CHILD ($p = 20$). BBB methods were executed with $n_0 = 3200$ using hybrid MCMC to estimate the structure posterior.

numerical experiments against standard MAB algorithms as well as a generously optimistic version of a recently proposed CB approach.

Although our work notably does not depend on certain restrictive assumptions made by previous work, namely knowledge of the causal graph or large-sample observational information, our proposed methodology nonetheless requires a causally sufficient system which may not be available in practice. In the presence of unobserved confounders, an obvious challenge is that the variables in the parent sets that BBB uses for backdoor adjustment may not all be observed. Since sets that satisfy the backdoor criterion are not limited to the parent set, one approach to this setting would be to otherwise identify valid adjustment sets. Perhaps the most obvious extension of our methodology would be to model the underlying ancestral graph instead of the DAG. From the ancestral graph, causal effects may be estimated from observational data by identifying valid backdoor adjustment sets based on its structure.

The challenge of scaling BBB was addressed briefly by hybrid MCMC in Section 6, but the limits of the computational feasibility of BBB have yet to be carefully investigated or precisely articulated. Preliminary investigations suggest that BBB can scale to well over 100 variables, but extended empirical evaluation is necessary. A careful technical study of the posterior distributions is left as future work to complete the theoretical analysis of the cumulative regret. Finally, it would be interesting to consider how to share information between arms in the discrete setting as in Proposition 3 with a similarly simple graphical criterion.

## Acknowledgements

This work was supported by US NSF grant DMS-1952929. This work used computational and storage services associated with the Hoffman2 Shared Cluster provided by UCLA Institute for Digital Research and Education's Research Technology Group.

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

# A  Proofs and Discussions

*Proof of Proposition 1.* Let $\mathcal{T}_a = \{t \leq T : a_t = a\}$ be the time steps in which $a$ is selected. Then by inequality (10), Assumption 1 and the concavity of $\sqrt{x}$, we have

$$\sum_{t \in \mathcal{T}_a} \mathrm{E}[U_a(t) - \mu_a] \leq c\sqrt{\log T} \left\{ c_1 \sum_{t \in \mathcal{T}_a} [n_0 + n_a(t-1)]^{-\frac{1}{2}} + c_2\sqrt{d_a} \sum_{t \in \mathcal{T}_a} \exp\left[ -\frac{\delta_a n_a(t-1)}{2} \right] \right\}.$$

Now the first summation on the right side

$$\sum_{t \in \mathcal{T}_a} [n_0 + n_a(t-1)]^{-\frac{1}{2}} = \sum_{j=0}^{n_a(T)-1} (n_0 + j)^{-\frac{1}{2}}$$

$$\leq \int_{n_0-1}^{n_0+n_a(T)-1} x^{-\frac{1}{2}} dx = 2(\sqrt{(n_0-1) + n_a(T)} - \sqrt{n_0-1}).$$

Similar derivation shows that the second summation

$$\sum_{t \in \mathcal{T}_a} \exp\left[ -\frac{\delta_a n_a(t-1)}{2} \right] \leq c_a,$$

where $c_a$ is a constant. Therefore,

$$\sum_{t \in \mathcal{T}_a} \mathrm{E}[U_a(t) - \mu_a] \leq c\left[ 2c_1(\sqrt{(n_0-1) + n_a(T)} - \sqrt{n_0-1}) + c_2 c_a \sqrt{d_a} \right] \sqrt{\log T}.$$

By Cauchy-Schwartz inequality,

$$\sum_{a \in \mathcal{A}} \sqrt{(n_0-1) + n_a(T)} \leq \sqrt{K} \left\{ K(n_0-1) + \sum_a n_a(T) \right\}^{1/2} = \sqrt{KT + K^2(n_0-1)}.$$

Summing over actions, we arrive at

$$\sum_{t=1}^{T} \mathrm{E}[U_a(t) - \mu_a] = \sum_{a \in \mathcal{A}} \sum_{t \in \mathcal{T}_a} \mathrm{E}[U_a(t) - \mu_a]$$

$$\leq \left[ c_4 \left( \sqrt{KT + K^2(n_0-1)} - \sqrt{K^2(n_0-1)} \right) + c_5 \sum_{a \in \mathcal{A}} \sqrt{d_a} \right] \sqrt{\log T}, \qquad (17)$$

where $c_4 = 2cc_1$ and $c_5 = cc_2 \max_a c_a$.

Next, we show that the second term in (9) is bounded. By the definition of $U_a(t)$ in (7), Assumption 2 implies

$$P(\mu^* > U_{a^*}(t)) = \mathrm{E}\left[ P(\mu_{a^*} > U_{a^*}(t) \mid \mathcal{D}[t-1]) \right] \leq c_3(t-1)^{-b}$$

for $t \geq 2$, and thus,

$$\sum_{t=1}^{T} P(\mu_{a^*} > U_{a^*}(t)) \leq 1 + c_3 \sum_{t=1}^{\infty} t^{-b}.$$

Therefore, the second term is bounded by a constant $c_6$. Accordingly, (11) follows from (17). □

*Discussion of Assumption 1 and Assumption 2.* We now demonstrate how to verify Assumption 1 and Assumption 2 for Proposition 1. Recall that the posterior distribution $p(\mu_a \mid \mathcal{D}[t])$ is defined via averaging over possible parent set $\mathbf{Z}_a \coloneqq \mathbf{Pa}_{\langle a \rangle}^{\mathcal{G}} \subseteq \mathbf{X} \setminus X_{\langle a \rangle}$.

We start with decomposing $\mathrm{Var}(\mu_a \mid \mathcal{D}[t])$ by conditioning on $\mathbf{Z}_a$,

$$\mathrm{Var}(\mu_a \mid \mathcal{D}[t]) = \mathrm{E}\left[\mathrm{Var}(\mu_a \mid \mathbf{Z}_a, \mathcal{D}[t]) \mid \mathcal{D}[t]\right] + \mathrm{Var}\left[\mathrm{E}(\mu_a \mid \mathbf{Z}_a, \mathcal{D}[t]) \mid \mathcal{D}[t]\right]. \tag{18}$$

Using the Gaussian setting as an example, under the conjugate prior (15), the conditional posterior $p[\mu_a \mid \mathbf{Z}_a, \mathcal{D}[t]]$ is a $t$-distribution with $(n_0 + n_a(t) - |\mathbf{Z}_a|)$ degrees of freedom and variance

$$\mathrm{Var}(\mu_a \mid \mathbf{Z}_a, \mathcal{D}[t]) = \frac{V_a^t(\mathbf{Z}_a)}{n_0 + n_a(t)},$$

where $V_a^t(\mathbf{Z}_a) = O_p(1)$ depends on $\mathcal{D}[t]$. Now the first term on the right side of (18) is expressed as

$$\mathrm{E}\left[\mathrm{Var}(\mu_a \mid \mathbf{Z}_a, \mathcal{D}[t]) \mid \mathcal{D}[t]\right] = \sum_{\mathbf{Z}_a} \frac{V_a^t(\mathbf{Z}_a)}{n_0 + n_a(t)} P(\mathbf{Z}_a \mid \mathcal{D}[t]). \tag{19}$$

Taking further expectation to average over $\mathcal{D}[t]$, we get

$$\mathrm{E}\left\{\mathrm{E}\left[\mathrm{Var}(\mu_a \mid \mathbf{Z}_a, \mathcal{D}[t]) \mid \mathcal{D}[t]\right]\right\} = \frac{1}{n_0 + n_a(t)} \mathrm{E}\left[\sum_{\mathbf{Z}_a} V_a^t(\mathbf{Z}_a) P(\mathbf{Z}_a \mid \mathcal{D}[t])\right] \leq \frac{c_1^2}{n_0 + n_a(t)}, \tag{20}$$

where $c_1^2$ is an upper bound for the expectation in the second step for all $a \in \mathcal{A}$.

Let $\mu_a^t(\mathbf{Z}_a) := \mathrm{E}\left(\mu_a \mid \mathbf{Z}_a, \mathcal{D}[t]\right)$ and $\mathcal{G}^*$ be the true causal DAG. Then,

$$\mathrm{Var}\left[\mu_a^t(\mathbf{Z}_a) \mid \mathcal{D}[t]\right] \leq \sum_{\mathbf{Z}_a \neq \mathbf{Pa}_{\langle a \rangle}^{\mathcal{G}^*}} \left[\mu_a^t(\mathbf{Z}_a) - \mu_a^t(\mathbf{Pa}_{\langle a \rangle}^{\mathcal{G}^*})\right]^2 P(\mathbf{Z}_a \mid \mathcal{D}[t])$$

$$\leq \max_{\mathbf{Z}_a} \left[\mu_a^t(\mathbf{Z}_a) - \mu_a^t(\mathbf{Pa}_{\langle a \rangle}^{\mathcal{G}^*})\right]^2 P(\mathbf{Z}_a \neq \mathbf{Pa}_{\langle a \rangle}^{\mathcal{G}^*} \mid \mathcal{D}[t])$$

$$\leq 4C^2 P(\mathbf{Z}_a \neq \mathbf{Pa}_{\langle a \rangle}^{\mathcal{G}^*} \mid \mathcal{D}[t]),$$

where the last inequality is due to the assumption that $\mu_a \in [-C, C]$ for all $a$. Based on asymptotic approximation (Schwarz, 1978; Haughton, 1988), the posterior probability $P(\mathbf{Z}_a \mid \mathcal{D}[t]) = O_p(\exp[-\delta(\mathbf{Z}_a)n_a(t)])$ for any $\mathbf{Z}_a \neq \mathbf{Pa}_{\langle a \rangle}^{\mathcal{G}^*}$ when $n_a(t)$ is large, where $\delta(\mathbf{Z}_a) > 0$ is a constant depending on $\mathbf{Z}_a$. Let $d_a$ be the number of candidate parent sets for $X_{\langle a \rangle}$ and $\delta_a = \min\{\delta(\mathbf{Z}_a) : \mathbf{Z}_a \neq \mathbf{Pa}_{\langle a \rangle}^{\mathcal{G}^*}\}$. Taking expectation, we arrive at

$$\mathrm{E}\left\{\mathrm{Var}\left[\mu_a^t(\mathbf{Z}_a) \mid \mathcal{D}[t]\right]\right\} \leq c_2^2 d_a \exp(-\delta_a n_a(t)), \tag{21}$$

for some positive constant $c_2$. Combining (20) and (21) leads to Assumption 1.

Put $\mathbf{Z}_* \equiv \mathbf{Z}_{a^*}$ and $n_{a^*}(t) \equiv n_*(t)$. To verify Assumption 2, we make use of concentration of the conditional posterior distribution $p(\mu^* \mid \mathbf{Z}_*, \mathcal{D}[t])$ and concentration of $\mathrm{E}\left(\mu^* \mid \mathbf{Z}_*, \mathcal{D}[t]\right)$. Define two events,

$$\mathcal{E}_{t,1} := \left\{\frac{\mu^* - \mathrm{E}(\mu^* \mid \mathbf{Z}_*, \mathcal{D}[t])}{\sqrt{\mathrm{E}\left[\mathrm{Var}(\mu^* \mid \mathbf{Z}_*, \mathcal{D}[t]) \mid \mathcal{D}[t]\right]}} > \frac{c}{2}\sqrt{\log t}\right\},$$

$$\mathcal{E}_{t,2} := \left\{\frac{\mathrm{E}(\mu^* \mid \mathbf{Z}_*, \mathcal{D}[t]) - \mathrm{E}(\mu^* \mid \mathcal{D}[t])}{\sqrt{\mathrm{Var}\left[\mathrm{E}(\mu^* \mid \mathbf{Z}_*, \mathcal{D}[t]) \mid \mathcal{D}[t]\right]}} > \frac{c}{2}\sqrt{\log t}\right\}.$$

By (18), $\mathrm{Var}(\mu^* \mid \mathcal{D}[t]) \geq \mathrm{E}\left[\mathrm{Var}(\mu^* \mid \mathbf{Z}_*, \mathcal{D}[t]) \mid \mathcal{D}[t]\right]$ and $\mathrm{Var}(\mu^* \mid \mathcal{D}[t]) \geq \mathrm{Var}\left[\mathrm{E}(\mu^* \mid \mathbf{Z}_*, \mathcal{D}[t]) \mid \mathcal{D}[t]\right]$. Then, we have

$$P\left\{\frac{\mu^* - \mathrm{E}(\mu^* \mid \mathcal{D}[t])}{\sqrt{\mathrm{Var}(\mu^* \mid \mathcal{D}[t])}} > c\sqrt{\log t} \mid \mathcal{D}[t]\right\} \leq P(\mathcal{E}_{t,1} \mid \mathcal{D}[t]) + P(\mathcal{E}_{t,2} \mid \mathcal{D}[t]).$$

For the first probability, we further condition on $\mathbf{Z}_*$:

$$P(\mathcal{E}_{t,1} \mid \mathcal{D}[t]) = \sum_{\mathbf{Z}_*} P(\mathcal{E}_{t,1} \mid \mathbf{Z}_*, \mathcal{D}[t]) P(\mathbf{Z}_* \mid \mathcal{D}[t]).$$

According to (19), for a fixed $\mathcal{D}[t]$, $\mathrm{Var}(\mu^* \mid \mathbf{Z}_*, \mathcal{D}[t]) = O_p(\mathrm{E}\left[\mathrm{Var}(\mu^* \mid \mathbf{Z}_*, \mathcal{D}[t]) \mid \mathcal{D}[t]\right]) = O_p(1/[n_0 + n_*(t)])$. Then for some constant $c(\mathbf{Z}_*) > 0$,

$$P(\mathcal{E}_{t,1} \mid \mathbf{Z}_*, \mathcal{D}[t]) \leq P\left\{ \frac{\mu^* - \mathrm{E}(\mu^* \mid \mathbf{Z}_*, \mathcal{D}[t])}{\sqrt{\mathrm{Var}(\mu^* \mid \mathbf{Z}_*, \mathcal{D}[t])}} > c(\mathbf{Z}_*)\sqrt{\log t} \, \Big| \, \mathbf{Z}_*, \mathcal{D}[t] \right\}.$$

Upper bounds for the right side can be established based on concentration of many common posterior distributions. For the Gaussian setting, $\mu^* \mid \mathbf{Z}_*, \mathcal{D}[t]$ follows a $t$ distribution with $(n_0 + n_*(t) - |\mathbf{Z}_*|)$ degrees of freedom. Existing concentration inequality for $t$ distribution with $r$ degrees of freedom, such as

$$P(|t_r| \geq x) \leq 2\exp(-x^2/4) + \exp(-r/16)$$

in Lemma 18 of Wang (2022), can be used to show that $P(\mathcal{E}_{t,1} \mid \mathbf{Z}_*, \mathcal{D}[t]) = O(t^{-b})$ for some $b$ and every candidate $\mathbf{Z}_*$ and therefore $P(\mathcal{E}_{t,1} \mid \mathcal{D}[t]) = O(t^{-b})$ for any $\mathcal{D}[t]$.

Note that $\mathrm{E}(\mu^* \mid \mathbf{Z}_*, \mathcal{D}[t]) = \mu_{a^*}^t(\mathbf{Z}_*)$ is a function of $\mathbf{Z}_*$ conditioning on $\mathcal{D}[t]$ and thus a discrete and bounded random variable. The second probability

$$P(\mathcal{E}_{t,2} \mid \mathcal{D}[t]) = P\left\{ \frac{\mu_{a^*}^t(\mathbf{Z}_*) - \mathrm{E}(\mu_{a^*}^t(\mathbf{Z}_*) \mid \mathcal{D}[t])}{\sqrt{\mathrm{Var}\left[\mu_{a^*}^t(\mathbf{Z}_*) \mid \mathcal{D}[t]\right]}} > \frac{c}{2}\sqrt{\log t} \, \Big| \, \mathcal{D}[t] \right\}$$

may be shown to be $O(t^{-b})$ by existing concentration inequality of a discrete bounded random variable and the concentration of $[\mathbf{Z}_* \mid \mathcal{D}[t]]$ on the true parent set.

*Proof of Proposition 2.* Let $\mathbf{Z}$ be the parent set of $X$. Then by a special case of (1),

$$p(y \mid do(x)) = \int p(y \mid x, \mathbf{z}) p(\mathbf{z}) d\mathbf{z}$$

$$= \int \phi(y \mid \psi x + \boldsymbol{\gamma}^\top \mathbf{z}, \sigma^2) \phi(\mathbf{z} \mid 0, \Sigma_{\mathbf{Z}}) d\mathbf{z}$$

$$= \phi(y \mid \psi x, \boldsymbol{\gamma}^\top \Sigma_{\mathbf{Z}} \boldsymbol{\gamma} + \sigma^2),$$

where $\phi(\cdot \mid \mu, \Sigma)$ is the probability density function of $\mathrm{N}(\mu, \Sigma)$ and $\Sigma_{\mathbf{Z}}$ is the covariance matrix of $\mathbf{Z}$. Thus,

$$Y \mid do(X = x) \sim \mathrm{N}(\psi x, \boldsymbol{\gamma}^\top \Sigma_{\mathbf{Z}} \boldsymbol{\gamma} + \sigma^2).$$

Now representing $[Y \mid X, \mathbf{Z}]$ by a linear regression:

$$Y = \psi X + \boldsymbol{\gamma}^\top \mathbf{Z} + \varepsilon,$$

where $\varepsilon \sim N(0, \sigma^2) \perp \mathbf{Z} \sim N(0, \Sigma_{\mathbf{Z}})$. Then we have

$$\mathrm{Var}_P(Y - \psi X) = \mathrm{Var}_P(\boldsymbol{\gamma}^\top \mathbf{Z} + \varepsilon)$$

$$= \boldsymbol{\gamma}^\top \Sigma_{\mathbf{Z}} \boldsymbol{\gamma} + \sigma^2 = \mathrm{Var}_P(Y \mid do(X = x)).$$

$\square$

*Proof of Proposition 3.* The result follows straightforwardly from a simple graphical argument. Let $\Xi_{XY}^{\mathcal{G}}$ denote the distinct directed paths from $X$ to $Y$ in the causal graph $\mathcal{G}$ given the model (13), where $\xi \in \Xi_{XY}^{\mathcal{G}}$

consists of all the directed edges $i \to j \in \mathbf{E}$ on the given path from $X$ to $Y$. Then the causal effect of $X$ on $Y$ can be expressed as the sum of propagated direct effects along all directed paths from $X$ to $Y$:

$$\psi_{XY} := \frac{\partial}{\partial x} \mathrm{E}_P[Y \mid do(X = x)] = \sum_{\xi \in \Xi_{XY}^{\mathcal{G}}} \prod_{i \to j \in \xi} \beta_{ij}.$$

We denote the variables under the intervention $do(W = w)$, $w \in \mathbb{R}$ as $\tilde{\mathbf{X}}$, with resulting causal model

$$\tilde{X}_j = \sum_{i=1}^{p} \tilde{\beta}_{ij} \tilde{X}_i + \tilde{\varepsilon}_j, \quad j = 1 \dots, p,$$

where

$$\tilde{\beta}_{ij} = \begin{cases} 0 & \text{if } X_j = W \\ \beta_{ij} & \text{otherwise,} \end{cases} \qquad \tilde{\varepsilon}_j = \begin{cases} w & \text{if } X_j = W \\ \varepsilon_j & \text{otherwise.} \end{cases}$$

The corresponding causal graph for $\tilde{\mathbf{X}}$ is the mutilated graph $\tilde{\mathcal{G}}$ resulting from deleting all edges into $W$. The causal effect of $\tilde{X}$ on $\tilde{Y}$ is then

$$\psi_{\tilde{X}\tilde{Y}} := \frac{\partial}{\partial x} \mathrm{E}_P[\tilde{Y} \mid do(\tilde{X} = x)] = \frac{\partial}{\partial x} \mathrm{E}_P[Y \mid do(X = x), do(W = w)] = \sum_{\xi \in \Xi_{XY}^{\tilde{\mathcal{G}}}} \prod_{i \to j \in \xi} \tilde{\beta}_{ij}.$$

Since $W$ does not block any directed path from $X$ to $Y$, the mutilated graph $\tilde{\mathcal{G}}$ retains all the directed paths from $X$ to $Y$ in $\mathcal{G}$, so $\Xi_{XY}^{\tilde{\mathcal{G}}} = \Xi_{XY}^{\mathcal{G}}$. By the same reasoning, $\tilde{\beta}_{ij} = \beta_{ij}$ for all $i \to j \in \xi$ where $\xi \in \Xi_{XY}^{\tilde{\mathcal{G}}}$. Therefore, for any $w \in \mathbb{R}$,

$$\frac{\partial}{\partial x} \mathrm{E}_P[Y \mid do(X = x)] = \frac{\partial}{\partial x} \mathrm{E}_P[Y \mid do(X = x), do(W = w)].$$

$\square$

## B  Experimental Details

In this section, we include details regarding the experiments discussed in Section 6.

For our random network simulations, we generated CBN models for $p = 10$ variables with reward variable $Y = X_p$. In order to investigate interesting structures with diverse non-trivial confounding relationships, we randomly generated graph structures using the following process adapted from de Kroon et al. (2022). Given a fixed topological sort of the variables $X_1 \prec \cdots \prec X_p$ where the reward variable is $Y = X_p$, we sequentially considered nodes in reverse topological order: $i = p - 1, \dots, 1$. We uniformly sampled the maximum out-degree of $X_i$, denoted $d_i$, from 1 to $p - i$. Then, for $d_i$ times, we randomly selected $X_j$ from $\{X_j \in \mathbf{X} : X_i \prec X_j\}$, adding $X_i \to X_j$ to the graph only if the edge was not already present and $|\mathbf{Pa}_j^{\mathcal{G}}| < 3$. We imposed an additional requirement that $|\mathbf{Pa}_p^{\mathcal{G}}| = 3$, randomly adding parents if necessary. If the generated structure consisted of multiple disconnected components, we rejected the structure and reattempted the process.

The conditional probability distributions of each CBN were likewise generated randomly. For discrete networks, the variables were all assumed to be binary, and the conditional probability tables were randomly generated uniformly and normalized, and were accepted only if for every edge $X_j \to X_i$, there is a sufficiently large causal effect, with $|P[X_i = x_i \mid do(X_j = x_j)] - P(X_i = x_i)| \geq 0.05$ for some $x_i \in \mathrm{Dom}(X_i)$ and $x_j \in \mathrm{Dom}(X_j)$. Additionally, we required the marginal probability of any single discrete level to be at least 0.01, and that the reward signal of the optimal intervention $a^*$ be sufficiently large with respect to the observational mean: $\mu^* - \mathrm{E}[Y] \geq 0.05$. For Gaussian networks, according to the model expressed in (13), we sampled coefficients uniformly from $[-1, -0.5] \cup [0.5, 1]$ for $X_i \in \mathbf{Pa}_j^{\mathcal{G}}$ and standard deviations from $[\sqrt{0.5}, 1]$, and we normalized the system to have unit variance. Note that in the Gaussian setting, there are

effectively $|\mathcal{A}| = 9$ actions given that interventional data on the same variable may be combined as discussed in Section 5.2, which we implement for the competing methods as well. We found that $\langle a^* \rangle \in \mathbf{Pa}_p^{\mathcal{G}}$ held for 98% of the discrete models that we randomly generated, though only for 65% of the random Gaussian models. As discussed in Section 6, we artificially enforced $\langle a^* \rangle \in \mathbf{Pa}_p^{\mathcal{G}}$ when evaluating the regret of TS* and (Bayes-)UCB*.

The randomly generated networks were limited to size $p = 10$ in the interest of extending the scope of our empirical investigation in other aspects, namely in representing a large number of random causal models and executing enough repetitions and time steps to reasonably assess the expected performance. We emphasize that this limitation is primarily due to the breadth of our simulation study, whereas in practical applications there may not be the need for tens of thousands of executions.

The in-degree restriction of $|\mathbf{Pa}_j^{\mathcal{G}}| \leq 3$ for all $j \in \mathbf{V}$ was largely due to the difficulty in reliably generating random conditional probability distributions that have meaningful causal effects and reward signals, as defined in the previous paragraph, for denser discrete networks. In general, it is not uncommon to assume the underlying DAG structure is sparse (Kalisch & Bühlmann, 2007). Similarly, the choice of $|\mathbf{Pa}_p^{\mathcal{G}}| = 3$ was motivated by our interest in investigating non-trivial structures that have substantive connectivity between the reward variable and the intervened variables. Note that without sufficiently meaningful connectivity and causal effects, our BBB methodology is actually advantaged in that the interventional distributions generally will not be substantively different than the observational distribution, thus nullifying the need for backdoor adjustment and correspondingly causal structure learning.

For Bayes-UCB(*), the best quantile constant in (8) was $c = 0$, in agreement with the empirical recommendation by Kaufmann et al. (2012). The best exploration parameter for UCB in (6) was $c = 1/(2\sqrt{2})$ for UCB(*) in the discrete setting. In the Gaussian setting, UCB and UCB* preferred $c = 1/2$ and $c = 1/\sqrt{2}$, respectively, the latter of which we applied for BBB. We used standard uninformative priors for TS(*), with $\alpha_0 = \beta_0 = 1$ for the Beta prior and $m_0 = 0$, $\nu_0 = 1$, and $u_0 = v_0 = 1$ for the N-$\Gamma^{-1}$ prior. For BBB, we computed exact parent set probabilities (4) using the program[1] implementing the efficient algorithm developed by and applied in Pensar et al. (2020), restricting the maximum size of parent sets to three and using the Bayesian Dirichlet equivalent uniform and Bayesian Gaussian equivalent scores. For the Gaussian setting, we checked the graphical criterion in Proposition 3 according to (16) with $\tau = 0.1$.

While we focused in Section 3 on designing the marginal posteriors according to (3), a notable difference between our proposed Bayesian CB framework and the Bayesian MAB approach described in Section 2 is that in our design, the posterior distribution is not modular, with the marginals $(\pi_a^t)_{a \in \mathcal{A}}$ mutually dependent on the distribution of graph structures. However, because of software limitations and for simplicity, we sampled the criterion $U_a(t)$ for each arm independently in the implementation of BBB-TS in our random network experiments (line 7 in Algorithm 1). Although preliminary results have shown the difference in empirical performance to be negligible, a more precise implementation would first sample a DAG $\mathcal{G}$ from the posterior distribution $P(\mathcal{G} \mid \mathcal{D}[t])$ and subsequently for each arm $a \in \mathcal{A}$, sample $U_a(t)$ from $\pi_{a \mid \mathbf{Pa}_{\langle a \rangle}^{\mathcal{G}}}^t$, which we apply using MCMC in our scaling experiments.

For our investigation of scaling BBB with MCMC, we used the same generated random networks as previously described for Figure 4. For the CHILD network, we coerced all variables to binary variables, with the extraneous discrete states removed by sequentially merging states with least marginal probability. We averaged the conditional probability distributions imposed by merged states weighted according to their marginal probabilities.

Order MCMC was implemented by extending the BiDAG package (Suter et al., 2021) to accommodate computing scores with ensemble data as described in Section 3. For each iteration of BBB, the structure posterior was estimated by conducting $10^4$ iterations with a thinning interval of 10 and discarding the first 20% as burn-in steps. The resulting set of DAGs were used for Bayesian model averaging in BBB-(Bayes-)UCB, and for BBB-TS one random DAG was selected. For hybrid MCMC, the search space was initially gently restricted by executing the PC algorithm (Spirtes & Glymour, 1991) with a relatively large threshold $\alpha = 0.1$ and only investigating conditioning sets of up to size one. For subsequent iterations, the search space

---

[1]Pensar et al. (2020) provided their code under the MIT License at `https://github.com/jopensar/BIDA`.

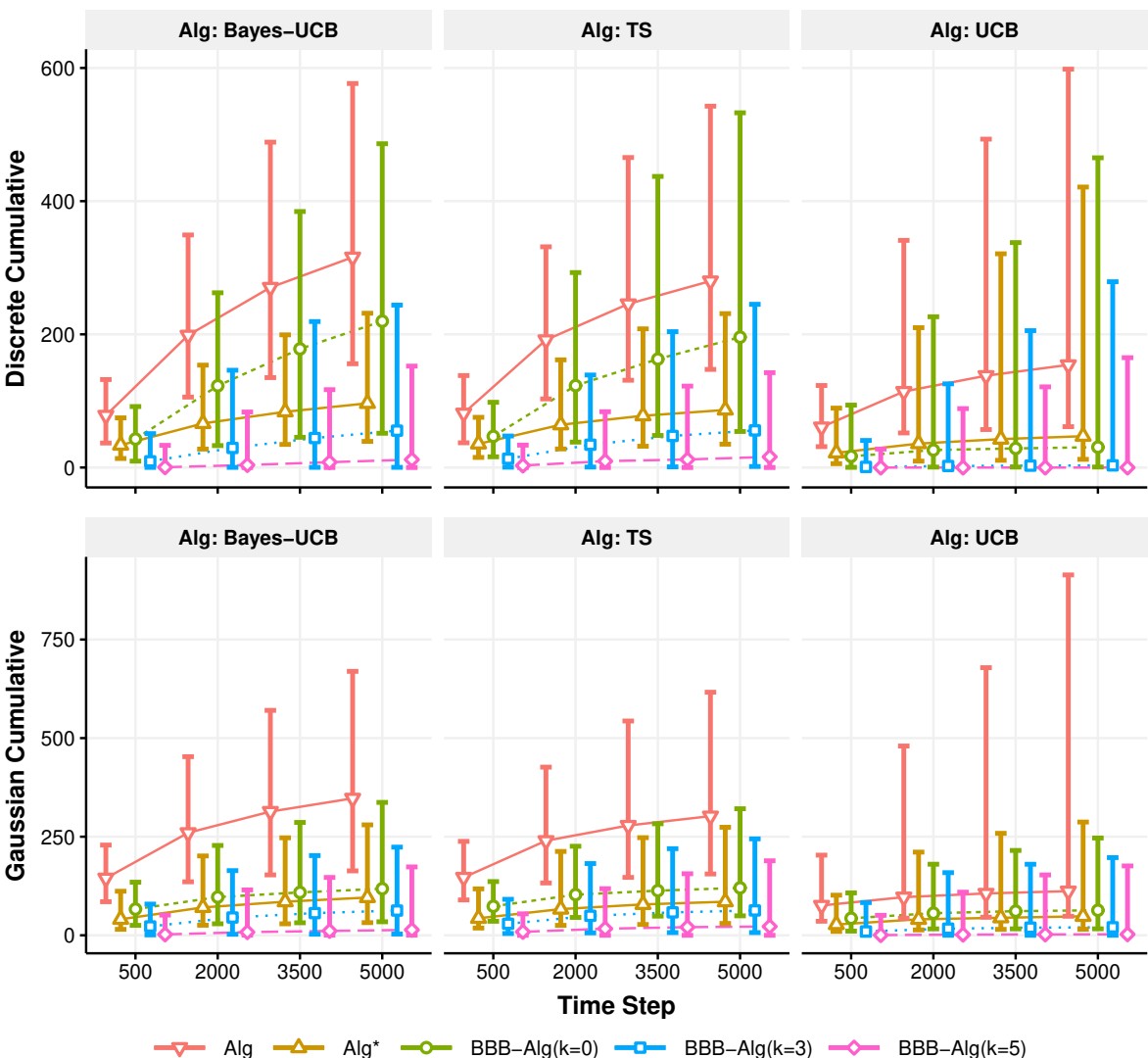

Figure 6: Median cumulative regret with 95% percentile error bars for time steps $t \in \{500, 2000, 3500, 5000\}$ comparing Alg, Alg*, and BBB-Alg for Alg $\in \{\text{Bayes-UCB}, \text{TS}, \text{UCB}\}$. BBB methods were executed with $n_0 = 100 \cdot 2^k$ in the discrete setting and $n_0 = 10 \cdot 2^k$ in the Gaussian setting.

was restricted to edges that appear with at least 0.05 probability in the structure posterior estimated in the preceding iteration. Note that the hybrid approach proposed by Kuipers et al. (2022) includes provisions for extending the search space for greater robustness in the presence of false negatives, so with each iteration the search space may be sequentially reduced or expanded as the structure posterior is increasingly informed by interventional data.

## C  Additional Results

Due to the density of information communicated in figures such as Figure 1, along with the substantial variability arising from the randomness in graph structures, conditional probability distributions, and data, we chose not to include error bounds of the empirical variability. To visualize the variability in the empirical results, we provide median cumulative regret with 95% percentile error bars in Figure 6.

Furthermore, in what follows we present the results from additional experiments designed to evaluate firstly our proposed approximation of the sampling variance of the discrete backdoor adjustment estimator (12), and

secondly the application of Proposition 3 by way of Gaussian backdoor adjustment with jointly interventional and observational data.

## C.1 Discrete Backdoor Adjustment and Variance

In this section, we describe and present experiments evaluating the behavior of $\hat{\mu}_{a,\text{bda}}(\mathbf{Z})$ where $\mathbf{Z} = \mathbf{Pa}^{\mathcal{G}}_{\langle a \rangle}$ as in (12), as well as our proposed approximation of its variance, derived in detail in Appendix D. Four variance estimation methods were investigated. In the naive approach, $\hat{\mu}_{a,\text{bda}}(\mathbf{Z})$ is treated as a conditional proportion as is the case when $|\mathbf{Z}| = 0$, and the variance is estimated as $\hat{\mu}_{a,\text{bda}}(\mathbf{Z})[1 - \hat{\mu}_{a,\text{bda}}(\mathbf{Z})]/n[x_a]$ where $n[x_a]$ is the number of samples of data where $X_{\langle a \rangle} = x_a$. The sampling approach estimates the variance from samples from the population distribution, and the bootstrap approach conducts resampling from each sample distribution, each with $10^3$ repetitions.

The generation of discrete CBNs for the simulation scenarios was designed as follows. The graph structure was generated simply by initializing a structure where there is a direct edge from the intervened node $X_{\langle a \rangle}$ to the reward variable $Y$ and $X_{\langle a \rangle}$ has $|\mathbf{Z}| = m$ parents. For each parent $X_j \in \mathbf{Z}$, an edge $X_j \to Y$ was randomly added with 0.5 probability to create backdoor paths. Finally, conditional probability tables were generated uniformly as described in Section 6.

For observational sample sizes $n_0 \in \{100 \cdot 2^k : k = 0, 1, \ldots, 5\}$ and parent set sizes $|\mathbf{Z}| \in \{0, 1, 2, 3\}$, $10^3$ scenarios were created by randomly generating CBNs as described above and the methods were assessed under each scenario through the following process. First, $10^6$ datasets were generated, each with $n_0$ samples of observational data, and for each dataset, $\hat{\mu}_{a,\text{bda}}(\mathbf{Z})$ was computed for some arbitrary $x_a \in \text{Dom}(X_{\langle a \rangle})$. Then, for each of the four methods, the variance was estimated corresponding to the first $10^3$ estimates of $\hat{\mu}_{a,\text{bda}}(\mathbf{Z})$, and from those the 2 standard deviation interval coverage probability of the true $\mu_a$ was computed.

The estimator $\hat{\mu}_{a,\text{bda}}(\mathbf{Z})$ itself was found to be generally unbiased, with the average of the $10^6$ estimates deviating from the true $\mu_a$ by less than 2% in over 99% of the 24,000 scenarios. The coverage probability results are shown in Figure 7, where each boxplot visualizes the coverage probability of a method across $10^3$ scenarios randomly generated under the given simulation setting. The outliers and invalid values, which typically corresponded to extreme scenarios, were removed. The naive approach is only correct when $|\mathbf{Z}| = 0$ and performs poorly when otherwise. The general results may be summarized as Naive < Bootstrap $\approx$ Proposed < Sampling, though our proposed estimator appears to outperform the bootstrap approach for larger $|\mathbf{Z}|$ and perform comparably with the population sampling approach for larger $n_0$ while requiring significantly less and nearly negligible computational expense compared to either.

## C.2 Gaussian Backdoor Adjustment with Ensemble Data

In this section, we empirically validate our methodology of conducting the regression (14) with jointly interventional and observational data to estimate $\psi_{\langle a \rangle}$, as discussed in Section 5.2. In particular, we compare the coverage probability of $\hat{\psi}_{\langle a \rangle,\text{bda}}(\mathbf{Z})$ where $\mathbf{Z} = \mathbf{Pa}^{\mathcal{G}}_{\langle a \rangle}$ estimated using purely observational data and ensemble data. The ensemble data was generated by allowing each data sample to be generated by one of the possible interventions $\{do(X_j = x_j) : X_j \in \mathbf{Z}, x_j \in \{-1, 1\}\}$ or by passive observation, with equal probability given to each of the $2|\mathbf{Z}| + 1$ options.

For sample sizes $n \in \{10 \cdot 2^k : k = 0, 1, \ldots, 5\}$ and parent set sizes $|\mathbf{Z}| \in \{1, 2, 3, 4\}$, $10^3$ scenarios were created by randomly generating CBNs. The network structures were generated as described in Appendix C.1, and the parameters as in Section 6. Each data generation method was evaluated for each scenario by generating $10^5$ datasets with $n$ samples and estimating $\hat{\psi}_{\langle a \rangle,\text{bda}}(\mathbf{Z})$ and $\hat{\text{SE}}^2[\hat{\psi}_{\langle a \rangle,\text{bda}}(\mathbf{Z})]$ for each dataset by conducting the regression (14). From those estimates, 95% confidence interval coverage probabilities were computed for each scenario.

The average of the $10^5$ estimates of $\hat{\psi}_{\langle a \rangle,\text{bda}}(\mathbf{Z})$ deviated from the true $\psi_{\langle a \rangle}$ by at most 0.9% across all 24,000 simulation scenarios for both data generation methods. The coverage probability results are shown in Figure 8. Since the results did not vary across parent set sizes, each boxplot visualizes the coverage probability of a method across the 4,000 simulation scenarios at each sample size. It is easy to see equivalent

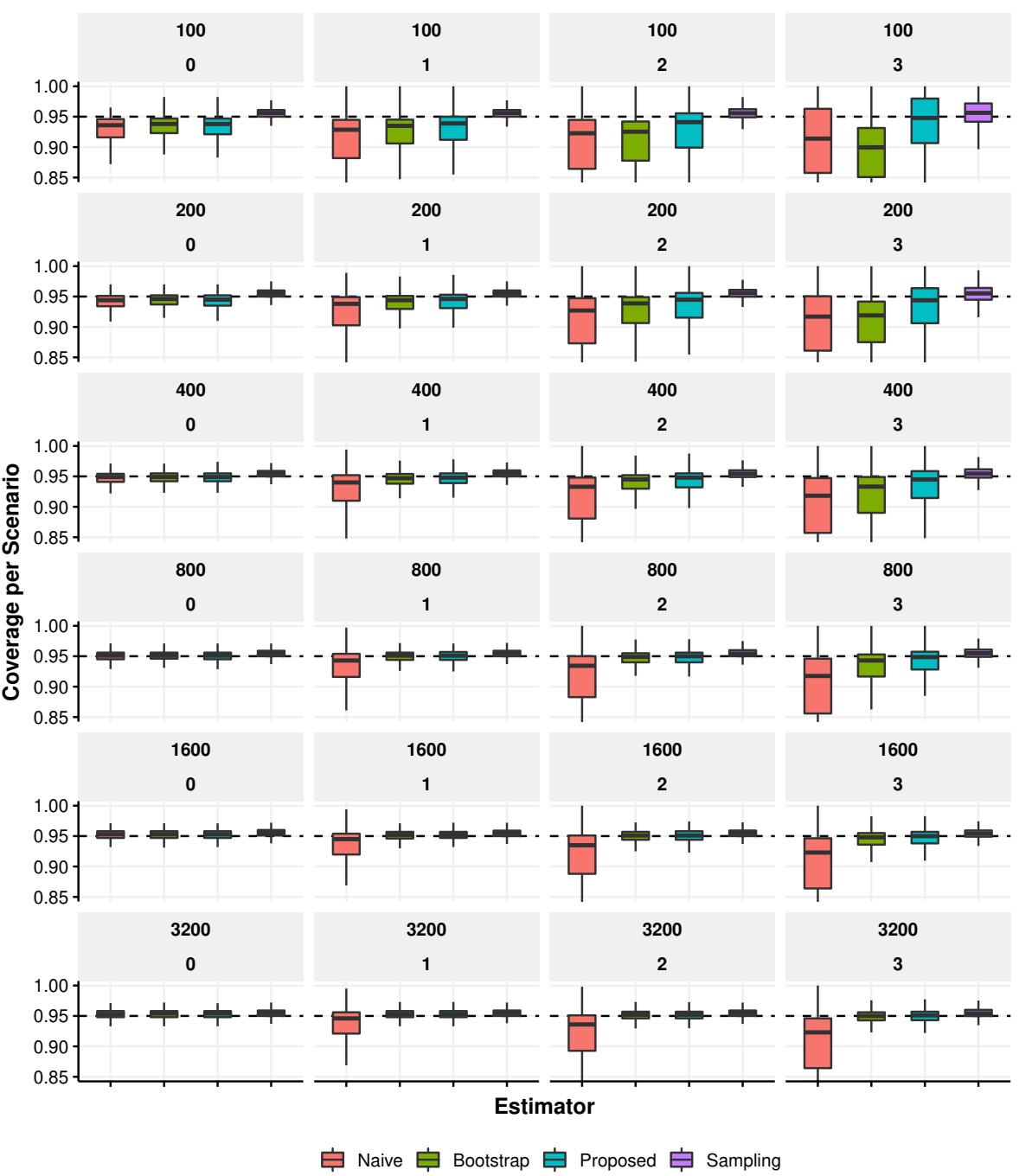

Figure 7: Coverage probability per scenario using various estimators of $\mathrm{Var}[\hat{\mu}_{a,\mathrm{bda}}(\mathbf{Z})]$ across $n_0 \in \{100 \cdot 2^k : k = 0, 1, \dots, 5\}$ samples of observational data and $|\mathbf{Z}| \in \{0, 1, 2, 3\}$ adjustment set sizes.

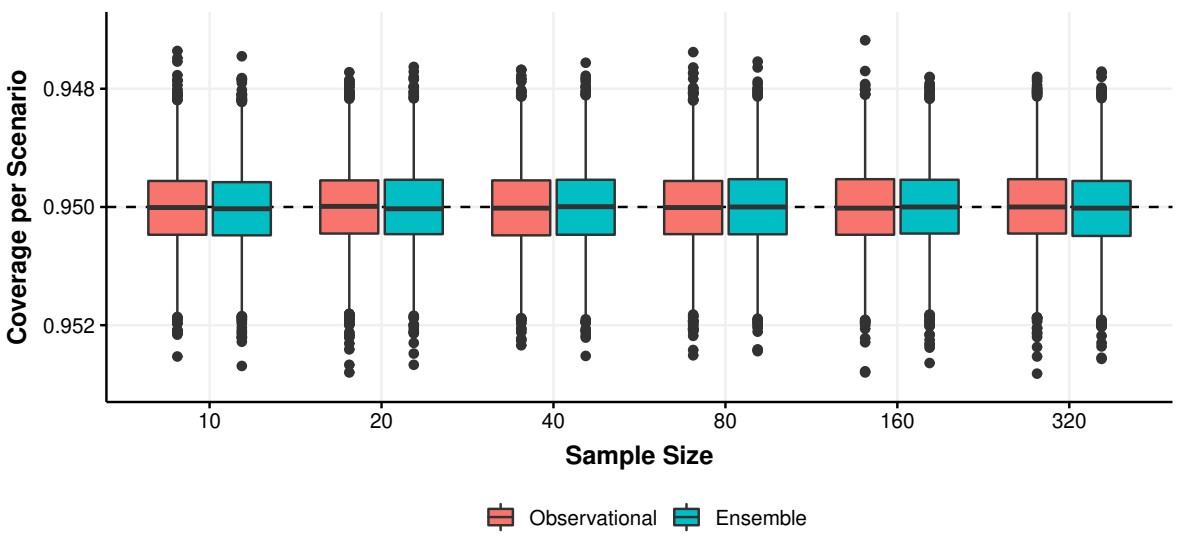

Figure 8: Coverage probability per simulation scenario across sample sizes for observational and ensemble data generating methods.

performance of the estimator computed with ensemble data compared to observational data, with consistent coverage across all sample sizes.

# D    Derivation of the Discrete Backdoor Adjustment Variance Approximation

In this section, we derive the approximation of the sampling variance of (12):

$$\hat{\mu}_{a,\text{bda}}(\mathbf{Z}) = \frac{1}{n_0} \sum_{\mathbf{z}} \frac{n_0[1, x_a, \mathbf{z}] n_0[\mathbf{z}]}{n_0[x_a, \mathbf{z}]}.$$

For the entirety of this section, we assume that the expectations and variances are with respect to the discrete probability distribution $P$ defined by a fixed CBN $\mathcal{B}$.

## D.1    Introduction

For simplicity, we redefine some notation. The backdoor adjustment to estimate the interventional distribution of $Y \mid do(X = x)$ with parent set $\mathbf{Z} = \mathbf{Pa}_X^{\mathcal{G}}$ with $r$ parent configurations is given by:

$$P[Y = y \mid do(X = x)] = \sum_{\mathbf{z}} P(Y = y \mid X = x, \mathbf{Z} = \mathbf{z}) P(\mathbf{Z} = \mathbf{z}).$$

Empirically, given $n$ samples of observational data, this quantity is estimated using counts:

$$\hat{P}[Y = y \mid do(X = x)] = \sum_{\mathbf{z}} \frac{n[y, x, \mathbf{z}]}{n[x, \mathbf{z}]} \frac{n[\mathbf{z}]}{n} = \frac{1}{n} \sum_{\mathbf{z}} \frac{n[y, x, \mathbf{z}] n[\mathbf{z}]}{n[x, \mathbf{z}]}, \tag{22}$$

where $n[y, x, \mathbf{z}]$ represents the number of samples in which $Y = y$, $X = x$, and $\mathbf{Z} = \mathbf{z}$, with corresponding definitions for $n[x, \mathbf{z}]$ and $n[\mathbf{z}]$. The joint probability distribution of $X$, $Y$, and $\mathbf{Z}$ may be lumped into a multinomial random vector $\mathbf{N} = (N_1, N_1{}', N_1{}'', \ldots, N_r, N_r{}', N_r{}'') \in \mathbb{R}^{3r}$ where for $i = 1, \ldots, r$,

$$N_i = n[y, x, \mathbf{z}_i], \quad N_i' = n[\neg y, x, \mathbf{z}_i], \quad N_i'' = n[\neg x, \mathbf{z}_i].$$

Note that $N_i + N_i' + N_i'' = n[\mathbf{z}_i]$, so $\sum_{i=1}^{r}(N_i + N_i' + N_i'') = n$, so $\mathbf{N}$ may be thought of as a repartitioning of the joint probability distribution of $X$, $Y$, and $\mathbf{Z}$ into $3r$ disjoint levels:

$$
\begin{aligned}
\mathbf{N} &= (N_1, N_1', N_1'', \ldots, N_r, N_r', N_r'') \sim \text{Multinom}(n, \mathbf{p}), \\
\mathbf{p} &= (p_1, p_1', p_1'', \ldots, p_r, p_r', p_r''), \quad \text{where} \\
p_i &= \text{E}\left[\frac{n[y, x, \mathbf{z}_i]}{n}\right], \quad p_i' = \text{E}\left[\frac{n[\neg y, x, \mathbf{z}_i]}{n}\right], \quad p_i'' = \text{E}\left[\frac{n[\neg x, \mathbf{z}_i]}{n}\right] \quad \text{for } i = 1, \ldots, r.
\end{aligned}
\tag{23}
$$

The advantage of such a representation is so that for each $\mathbf{z}_i$, the term within the summation may be expressed as a function of three disjoint elements of a multinomial random vector:

$$
\begin{aligned}
\frac{1}{n}\sum_{i=1}^{r}\frac{n[y, x, \mathbf{z}_i]n[\mathbf{z}_i]}{n[x, \mathbf{z}_i]} &= \frac{1}{n}\sum_{i=1}^{r}\frac{n[y, x, \mathbf{z}_i]\left(n[y, x, \mathbf{z}_i] + n[\neg y, x, \mathbf{z}_i] + n[\neg x, \mathbf{z}_i]\right)}{n[y, x, \mathbf{z}_i] + n[\neg y, x, \mathbf{z}_i]} \\
&= \frac{1}{n}\sum_{i=1}^{r}\frac{N_i(N_i + N_i' + N_i'')}{N_i + N_i'}.
\end{aligned}
\tag{24}
$$

Note that each term is not straightforward to compute. An obvious challenge is that the denominator of each term in the summation in (24) can be zero, so there is no analytical solution for its mean, variance, and covariance.

## D.2 Taylor Series Expansion for Ratio Distribution

To circumvent this challenge, we approximate the ratio in (24) with the Taylor series approximation. We begin by defining

$$
\begin{aligned}
M_i &= \frac{N_i(N_i + N_i' + N_i'')}{n^2}, \\
W_i &= \frac{N_i + N_i'}{n}, \\
Q_i &= f(M_i, W_i) = \frac{M_i}{W_i}.
\end{aligned}
$$

This allows us to express the variance of (24) in terms of $Q_i$:

$$
\begin{aligned}
\text{Var}\left[\hat{P}[Y = y \mid do(X = x)]\right] &= \text{Var}\left[\sum_{i=1}^{r}Q_i\right] \\
&= \sum_{i}\text{Var}\left[Q_i\right] + 2\sum_{i=1}^{r}\sum_{j>i}\text{Cov}\left[Q_i, Q_j\right].
\end{aligned}
\tag{25}
$$

By Taylor series expansion around $\mu_i = (\mu_{M_i}, \mu_{W_i}) = (\text{E}[M_i], \text{E}[W_i])$:

$$
\begin{aligned}
Q_i &= f(M_i, W_i) \\
&= f(\mu_i) + (M_i - \mu_{M_i})\frac{\partial f}{\partial M_i}(\mu_i) + (W_i - \mu_{W_i})\frac{\partial f}{\partial W_i}(\mu_i) \\
&\quad + \frac{1}{2}(M_i - \mu_{M_i})^2\frac{\partial^2 f}{\partial M_i^2}(\mu_i) + \frac{1}{2}(W_i - \mu_{W_i})^2\frac{\partial^2 f}{\partial W_i^2}(\mu_i) \\
&\quad + (M_i - \mu_{M_i})(W_i - \mu_{W_i})\frac{\partial^2 f}{\partial M_i \partial W_i}(\mu_i) \\
&\quad + O\left(\|(M_i, W_i) - \mu_i\|^3\right),
\end{aligned}
\tag{26}
$$

where

$$\frac{\partial f}{\partial M_i}(M_i, W_i) = \frac{1}{W_i}, \qquad \frac{\partial^2 f}{\partial M_i^2}(M_i, W_i) = 0,$$

$$\frac{\partial f}{\partial W_i}(M_i, W_i) = -\frac{M_i}{W_i^2}, \qquad \frac{\partial^2 f}{\partial W_i^2}(M_i, W_i) = \frac{2M_i}{W_i^3}, \qquad (27)$$

$$\frac{\partial^2 f}{\partial M_i \partial W_i}(M_i, W_i) = \frac{\partial^2 f}{\partial W_i \partial M_i}(M_i, W_i) = \frac{1}{W_i^2}$$

Given (26), we obtain an approximate expected value:

$$\mathrm{E}[Q_i] \approx f(\mu_i) + \frac{1}{2}\frac{\partial^2 f}{\partial M_i^2}(\mu_i)\mathrm{Var}[M_i] + \frac{1}{2}\frac{\partial^2 f}{\partial W_i^2}(\mu_i)\mathrm{Var}[W_i] + \frac{\partial^2 f}{\partial M_i \partial W_i}(\mu_i)\mathrm{Cov}[M_i, W_i]. \qquad (28)$$

For variance and covariance, we use a simpler approximation:

$$Q_i = f(M_i, W_i) \approx f(\mu_i) + (M_i - \mu_{M_i})\frac{\partial f}{\partial M_i}(\mu_i) + (W_i - \mu_{W_i})\frac{\partial f}{\partial W_i}(\mu_i), \qquad (29)$$

resulting in

$$\mathrm{Var}[Q_i] \approx \frac{\partial f}{\partial M_i}(\mu_i)^2 \mathrm{Var}[M_i] + \frac{\partial f}{\partial W_i}(\mu_i)^2 \mathrm{Var}[W_i]$$

$$+ 2\frac{\partial f}{\partial M_i}(\mu_i)\frac{\partial f}{\partial W_i}(\mu_i)\mathrm{Cov}[M_i, W_i], \qquad (30)$$

and

$$\mathrm{E}[Q_i Q_j] \approx f(\mu_i)f(\mu_j)$$

$$+ \frac{\partial f}{\partial M_i}(\mu_i)\frac{\partial f}{\partial M_j}(\mu_j)\mathrm{Cov}[M_i, M_j] + \frac{\partial f}{\partial M_i}(\mu_i)\frac{\partial f}{\partial W_j}(\mu_j)\mathrm{Cov}[M_i, W_j]$$

$$+ \frac{\partial f}{\partial W_i}(\mu_i)\frac{\partial f}{\partial M_j}(\mu_j)\mathrm{Cov}[W_i, M_j] + \frac{\partial f}{\partial W_i}(\mu_i)\frac{\partial f}{\partial W_j}(\mu_j)\mathrm{Cov}[W_i, W_j],$$

so

$$\mathrm{Cov}[Q_i, Q_j] = \mathrm{E}[Q_i Q_j] - \mathrm{E}[Q_i]\mathrm{E}[Q_j]$$

$$= \frac{\partial f}{\partial M_i}(\mu_i)\frac{\partial f}{\partial M_j}(\mu_j)\mathrm{Cov}[M_i, M_j] + \frac{\partial f}{\partial M_i}(\mu_i)\frac{\partial f}{\partial W_j}(\mu_j)\mathrm{Cov}[M_i, W_j] \qquad (31)$$

$$+ \frac{\partial f}{\partial W_i}(\mu_i)\frac{\partial f}{\partial M_j}(\mu_j)\mathrm{Cov}[W_i, M_j] + \frac{\partial f}{\partial W_i}(\mu_i)\frac{\partial f}{\partial W_j}(\mu_j)\mathrm{Cov}[W_i, W_j].$$

In what follows, we first derive important quantities from the multinomial distribution in Appendix D.3 and apply them to compute the quantities in (25).

### D.3 Multinomial Derivations

For this subsection, in an abuse of notation, let $\mathbf{N} = (N_1, \ldots, N_r) \sim \mathrm{Multinom}(n, \mathbf{p})$ and $u, v, w, x \in \{1, \ldots, r\}$ are distinct values. It is well-known that $\mathrm{E}[N_u] = np_u$, $\mathrm{Var}[N_u] = np_u(1-p_u)$, and $\mathrm{Cov}(N_u, N_v) = -np_u p_v$. Furthermore,

$$\mathrm{E}[N_u N_v] = \mathrm{Cov}[N_u, N_v] + \mathrm{E}[N_u]\mathrm{E}[N_v]$$

$$= n(n-1)p_u p_v, \qquad (32)$$

and the first four moments from derivating the moment generating function are:

$$\begin{aligned}
\mathrm{E}[N_u] &= np_u, \\
\mathrm{E}[N_u^2] &= n(n-1)p_u^2 + \mathrm{E}[N_u] \\
&= np_u[1 + (n-1)p_u], \\
\mathrm{E}[N_u^3] &= n(n-1)[(n-2)p_u^3 + 2p_u^2] + \mathrm{E}[N_u^2] \\
&= np_u\left[1 + (n-1)p_u(3 + (n-2)p_u)\right], \\
\mathrm{E}[N_u^4] &= n(n-1)(n-2)\left[(n-3)p_u^4 + 3p_u^3\right] + 2n(n-1)\left[(n-2)p_u^3 + 2p_u^2\right] + \mathrm{E}[N_u^3] \\
&= np_u\left[1 + (n-1)p_u(7 + (n-2)p_u[6 + (n-3)p_u])\right].
\end{aligned} \tag{33}$$

Define indicator random variable $U_i$ such that $U_i = 1$ if the outcome for trial $i$ is $u \in \{1, \ldots, r\}$ and $U_i = 0$ otherwise. Similarly define $V_i$ for $v \neq u$, $W_i$ for $w \neq v \neq u$, and $X_i$ for $x \neq w \neq v \neq u$. Then $N_u$, $N_v$, $N_w$, and $N_x$ may be expressed as

$$N_u = \sum_{i=1}^{n} U_i, \quad N_v = \sum_{i=1}^{n} V_i, \quad N_w = \sum_{i=1}^{n} W_i, \quad N_x = \sum_{i=1}^{n} X_i.$$

We are interested in $\mathrm{E}[N_u^2 N_v^2]$, $\mathrm{E}[N_u^3 N_v]$, $\mathrm{E}[N_u^2 N_v N_w]$, $\mathrm{E}[N_u N_v N_w N_x]$, $\mathrm{E}[N_u^2 N_v]$, and $\mathrm{E}[N_u N_v N_w]$.

$$\begin{aligned}
\mathrm{E}[N_u^2 N_v^2] &= \mathrm{E}\left[\left(\sum_{i=1}^{n} U_i\right)^2 \left(\sum_{i=1}^{n} V_i\right)^2\right] & \\
&= \mathrm{E}\left[\sum_{i=1}^{n}\sum_{j=1}^{n}\sum_{k=1}^{n}\sum_{l=1}^{n} U_i U_j V_k V_l\right] & \text{by distributing} \\
&= \sum_{i=1}^{n}\sum_{j=1}^{n}\sum_{k=1}^{n}\sum_{l=1}^{n} \mathrm{E}\left[U_i U_j V_k V_l\right] & \text{by linearity of expectation} \\
&= \sum_{i=1}^{n}\sum_{j=1}^{n}\sum_{\substack{k\neq i \\ k\neq j}}\sum_{\substack{l\neq i \\ l\neq j}} \mathrm{E}\left[U_i U_j V_k V_l\right] & \text{since } U_i V_i = 0 \text{ for all } i = 1, \ldots, n \\
&= \sum_{i=1}^{n}\sum_{j=1}^{n}\sum_{\substack{k\neq i \\ k\neq j}}\sum_{\substack{l\neq i \\ l\neq j}} \mathrm{E}\left[U_i U_j\right]\mathrm{E}\left[V_k V_l\right] & \text{by independence between trials} \\
&= \sum_{i=j}\sum_{\substack{k=l \\ k\neq i}} \mathrm{E}\left[U_i U_j\right]\mathrm{E}\left[V_k V_l\right] + \sum_{i}\sum_{j\neq i}\sum_{\substack{k\neq i \\ k\neq j}}\sum_{\substack{l\neq k \\ l\neq i \\ l\neq j}} \mathrm{E}\left[U_i U_j\right]\mathrm{E}\left[V_k V_l\right] & \\
&\quad + \sum_{i=j}\sum_{k\neq i}\sum_{\substack{l\neq k \\ l\neq i}} \mathrm{E}\left[U_i U_j\right]\mathrm{E}\left[V_k V_l\right] + \sum_{k=l}\sum_{i\neq k}\sum_{\substack{j\neq i \\ j\neq k}} \mathrm{E}\left[U_i U_j\right]\mathrm{E}\left[V_k V_l\right] & \text{reexpressed} \\
&= \sum_{i}\sum_{\substack{k=l \\ k\neq i}} \mathrm{E}[U_i^2]\mathrm{E}[V_k^2] + \sum_{i}\sum_{j\neq i}\sum_{\substack{k\neq i \\ k\neq j}}\sum_{\substack{l\neq k \\ l\neq i \\ l\neq j}} \mathrm{E}[U_i]\mathrm{E}[U_j]\mathrm{E}[V_k]\mathrm{E}[V_l] & \text{reexpressed; independence; and} \\
&\quad + \sum_{i}\sum_{k\neq i}\sum_{\substack{l\neq k \\ l\neq i}} \mathrm{E}[U_i^2]\mathrm{E}[V_k V_l] + \sum_{k}\sum_{i\neq k}\sum_{\substack{j\neq i \\ j\neq k}} \mathrm{E}[U_i]\mathrm{E}[U_j]\mathrm{E}[V_k^2] & \text{since } \mathrm{E}[U_i U_j] = \mathrm{E}[U_i]\mathrm{E}[U_j], \ i \neq j \\
&= n(n-1)p_u p_v + n(n-1)(n-2)(n-3)p_u^2 p_v^2 & \text{since } \mathrm{E}[U_i^2] = \mathrm{E}[U_i] = p_u \\
&\quad + n(n-1)(n-2)p_u p_v^2 + n(n-1)(n-2)p_u^2 p_v & \\
&= n(n-1)p_u p_v \left[1 + (n-2)(p_u + p_v + (n-3)p_u p_v)\right] & \text{simplified.}
\end{aligned}$$

Hence,

$$\mathrm{E}[N_u^2 N_v^2] = n(n-1)p_u p_v \left[1 + (n-2)(p_u + p_v + (n-3)p_u p_v)\right]. \tag{34}$$

Following the same derivation strategy,

$$E[N_u^3 N_v] = n(n-1)p_u p_v \left[1 + (n-2)p_u(3 + (n-3)p_u)\right], \tag{35}$$

$$E[N_u^2 N_v N_w] = n(n-1)(n-2)p_u p_v p_w \left[1 + (n-3)p_u\right], \tag{36}$$

$$E[N_u N_v N_w N_x] = n(n-1)(n-2)(n-3)p_u p_v p_w p_x, \tag{37}$$

$$E[N_u^2 N_v] = n(n-1)p_u p_v \left[1 + (n-2)p_u\right], \tag{38}$$

$$E[N_u N_v N_w] = n(n-1)(n-2)p_u p_v p_w. \tag{39}$$

### D.4 Numerator and Denominator of Ratio

We now turn to the task of deriving expressions for $\mathrm{Var}[M_i]$, $\mathrm{Var}[W_i]$, and $\mathrm{Cov}[M_i, W_i]$ in order to compute (30), and additionally for $\mathrm{Cov}[M_i, M_j]$, $\mathrm{Cov}[M_i, W_j]$, $\mathrm{Cov}[W_i, M_j]$, and $\mathrm{Cov}[W_i, W_j]$ for (31). For this subsection, return to the notation for $\mathbf{N}$ expressed in (23).

The distribution of $W_i = n^{-1}(N_i + N_i')$ is most simple. By the lumping property of multinomial random vectors,

$$\begin{aligned} E[W_i] &= p_i + p_i', \\ \mathrm{Var}[W_i] &= \frac{(p_i + p_i')(1 - p_i - p_i')}{n}, \\ \mathrm{Cov}[W_i, W_j] &= -\frac{(p_i + p_i')(p_j + p_j')}{n}. \end{aligned} \tag{40}$$

The distribution of $M_i = n^{-2}N_i(N_i + N_i' + N_i'')$ is more challenging. From (33) and (32), the expectation is given by:

$$\begin{aligned} E[M_i] &= n^{-2}E[N_i(N_i + N_i' + N_i'')] \\ &= n^{-2}\left(E[N_i^2] + E[N_i N_i'] + E[N_i N_i'']\right) \\ &= n^{-2}\left(np_i[1 + (n-1)p_i] + n(n-1)p_i p_i' + n(n-1)p_i p_i''\right) \\ &= n^{-1}p_i[1 + (n-1)(p_i + p_i' + p_i'')]. \end{aligned} \tag{41}$$

Next, the variance is given by:

$$\begin{aligned} \mathrm{Var}[M_i] &= n^{-4}\mathrm{Var}[N_i(N_i + N_i' + N_i'')] \\ &= n^{-4}\mathrm{Var}[N_i^2 + N_i N_i' + N_i N_i''] \\ &= n^{-4}\big(\mathrm{Var}[N_i^2] + \mathrm{Var}[N_i N_i'] + \mathrm{Var}[N_i N_i''] \\ &\qquad + 2\mathrm{Cov}[N_i^2, N_i N_i'] + 2\mathrm{Cov}[N_i^2, N_i N_i''] + 2\mathrm{Cov}[N_i N_i', N_i N_i'']\big). \end{aligned}$$

The terms in the expression above are given below. From the moments of the multinomial distribution (33):

$$\begin{aligned} \mathrm{Var}[N_i^2] &= E[N_i^4] - E[N_i^2]^2 \\ &= np_i\left[1 + (n-1)p_i(7 + (n-2)p_i[6 + (n-3)p_i])\right] - (np_i[1 + (n-1)p_i])^2 \\ &= np_i\left[1 + (n-1)p_i(7 + (n-2)p_i[6 + (n-3)p_i]) - np_i(1 + (n-1)p_i)^2\right]. \end{aligned}$$

From (34) and (32):

$$\begin{aligned} \mathrm{Var}[N_i N_i'] &= E[N_i^2 N_i'^2] - E[N_i N_i']^2 \\ &= n(n-1)p_i p_i'[1 + (n-2)(p_i + p_i' + (n-3)p_i p_i')] - [n(n-1)p_i p_i']^2 \\ &= n(n-1)p_i p_i'[1 + (n-2)(p_i + p_i' + (n-3)p_i p_i') - n(n-1)p_i p_i'], \\ \mathrm{Var}[N_i N_i''] &= n(n-1)p_i p_i''[1 + (n-2)(p_i + p_i'' + (n-3)p_i p_i'') - n(n-1)p_i p_i'']. \end{aligned}$$

From (35), (33), and (32):

$$
\begin{aligned}
\mathrm{Cov}[N_i^2, N_i N_i{}'] &= \mathrm{E}[N_i^3 N_i{}'] - \mathrm{E}[N_i^2]\mathrm{E}[N_i N_i{}'] \\
&= n(n-1)p_i p_i{}'\left[1 + (n-2)p_i(3 + (n-3)p_i)\right] \\
&\quad - np_i[1 + (n-1)p_i]n(n-1)p_i p_i{}' \\
&= n(n-1)p_i p_i{}'\left[1 + (n-2)(3p_i + (n-3)p_i^2) - np_i(1 + (n-1)p_i)\right] \\
\mathrm{Cov}[N_i^2, N_i N_i{}''] &= n(n-1)p_i p_i{}''\left[1 + (n-2)(3p_i + (n-3)p_i^2) - np_i(1 + (n-1)p_i)\right].
\end{aligned}
$$

From (36) and (32):

$$
\begin{aligned}
\mathrm{Cov}[N_i N_i{}', N_i N_i{}''] &= \mathrm{E}[N_i^2 N_i{}' N_i{}''] - \mathrm{E}[N_i N_i{}']\mathrm{E}[N_i N_i{}''] \\
&= n(n-1)(n-2)p_i p_i{}' p_i{}''[1 + (n-3)p_i] - n(n-1)p_i p_i{}' n(n-1)p_i p_i{}'' \\
&= n(n-1)p_i p_i{}' p_i{}''\left[(n-2)[1 + (n-3)p_i] - n(n-1)p_i\right].
\end{aligned}
$$

Hence, $\mathrm{Var}[M_i]$ is derived:

$$
\begin{aligned}
\mathrm{Var}[M_i] = n^{-4}\Big( & np_i\left[1 + (n-1)p_i(7 + (n-2)p_i[6 + (n-3)p_i]) - np_i(1 + (n-1)p_i)^2\right] \\
& + n(n-1)p_i p_i{}'[1 + (n-2)(p_i + p_i{}' + (n-3)p_i p_i{}') - n(n-1)p_i p_i{}'] \\
& + n(n-1)p_i p_i{}''[1 + (n-2)(p_i + p_i{}'' + (n-3)p_i p_i{}'') - n(n-1)p_i p_i{}''] \\
& + 2n(n-1)p_i p_i{}'\left[1 + (n-2)(3p_i + (n-3)p_i^2) - np_i(1 + (n-1)p_i)\right] \\
& + 2n(n-1)p_i p_i{}''\left[1 + (n-2)(3p_i + (n-3)p_i^2) - np_i(1 + (n-1)p_i)\right] \\
& + n(n-1)p_i p_i{}' p_i{}''\left[(n-2)[1 + (n-3)p_i] - n(n-1)p_i\right]\Big).
\end{aligned} \tag{42}
$$

Next, consider $\mathrm{Cov}[M_i, M_j]$.

$$
\begin{aligned}
\mathrm{Cov}[M_i, M_j] &= n^{-4}\mathrm{Cov}\left[N_i(N_i + N_i{}' + N_i{}''), N_j(N_j + N_j{}' + N_j{}'')\right] \\
&= n^{-4}\mathrm{Cov}\left[N_i^2 + N_i N_i{}' + N_i N_i{}'', N_j^2 + N_j N_j{}' + N_j N_j{}''\right] \\
&= n^{-4}\big(\mathrm{Cov}[N_i^2, N_j^2] \\
&\quad + \mathrm{Cov}[N_i^2, N_j N_j{}'] + \mathrm{Cov}[N_i^2, N_j N_j{}''] + \mathrm{Cov}[N_i N_i{}', N_j^2] + \mathrm{Cov}[N_i N_i{}'', N_j^2] \\
&\quad + \mathrm{Cov}[N_i N_i{}', N_j N_j{}'] + \mathrm{Cov}[N_i N_i{}', N_j N_j{}''] \\
&\quad + \mathrm{Cov}[N_i N_i{}'', N_j N_j{}'] + \mathrm{Cov}[N_i N_i{}'', N_j N_j{}'']\big).
\end{aligned}
$$

The terms in the expression above are given below. From (34) and (33):

$$
\begin{aligned}
\mathrm{Cov}[N_i^2, N_j^2] &= \mathrm{E}[N_i^2 N_j^2] - \mathrm{E}[N_i^2]\mathrm{E}[N_j^2] \\
&= n(n-1)p_i p_j\left[1 + (n-2)(p_i + p_j + (n-3)p_i p_j)\right] \\
&\quad - np_i[1 + (n-1)p_i]np_j[1 + (n-1)p_j] \\
&= np_i p_j[(n-1)(1 + (n-2)(p_i + p_j + (n-3)p_i p_j)) \\
&\quad - n(1 + (n-1)p_i)(1 + (n-1)p_j)].
\end{aligned}
$$

From (36), (33), and (32):

$$
\begin{aligned}
\mathrm{Cov}[N_i^2, N_j N_j{}'] &= \mathrm{E}[N_i^2 N_j N_j{}'] - \mathrm{E}[N_i^2]\mathrm{E}[N_j N_j{}'] \\
&= n(n-1)(n-2)p_i p_j p_j{}'[1 + (n-3)p_i] - np_i(1 + (n-1)p_i)n(n-1)p_j p_j{}' \\
&= n(n-1)p_i p_j p_j{}'\left[(n-2)[1 + (n-3)p_i] - n(1 + (n-1)p_i)\right], \\
\mathrm{Cov}[N_i^2, N_j N_j{}''] &= n(n-1)p_i p_j p_j{}''\left[(n-2)[1 + (n-3)p_i] - n(1 + (n-1)p_i)\right], \\
\mathrm{Cov}[N_i N_i{}', N_j^2] &= n(n-1)p_j p_i p_i{}'\left[(n-2)[1 + (n-3)p_j] - n(1 + (n-1)p_j)\right], \\
\mathrm{Cov}[N_i N_i{}'', N_j^2] &= n(n-1)p_j p_i p_i{}''\left[(n-2)[1 + (n-3)p_j] - n(1 + (n-1)p_j)\right].
\end{aligned}
$$

From (37) and (32):

$$
\begin{aligned}
\mathrm{Cov}[N_i N_i{}', N_j N_j{}'] &= \mathrm{E}[N_i N_i{}' N_j N_j{}'] - \mathrm{E}[N_i N_i{}']\mathrm{E}[N_j N_j{}'] \\
&= n(n-1)(n-2)(n-3)p_i p_i{}' p_j p_j{}' - n(n-1)p_i p_i{}' n(n-1)p_j p_j{}' \\
&= n(n-1)p_i p_i{}' p_j p_j{}' \left[(n-2)(n-3) - n(n-1)\right], \\
\mathrm{Cov}[N_i N_i{}', N_j N_j{}''] &= n(n-1)p_i p_i{}' p_j p_j{}'' \left[(n-2)(n-3) - n(n-1)\right], \\
\mathrm{Cov}[N_i N_i{}'', N_j N_j{}'] &= n(n-1)p_i p_i{}'' p_j p_j{}' \left[(n-2)(n-3) - n(n-1)\right], \\
\mathrm{Cov}[N_i N_i{}'', N_j N_j{}''] &= n(n-1)p_i p_i{}'' p_j p_j{}'' \left[(n-2)(n-3) - n(n-1)\right].
\end{aligned}
$$

Hence, $\mathrm{Cov}[M_i, M_j]$ is derived:

$$
\begin{aligned}
\mathrm{Cov}&[M_i, M_j] \\
&= n^{-4}\big(np_i p_j \big[(n-1)(1+(n-2)(p_i + p_j + (n-3)p_i p_j)) \\
&\qquad\qquad - n(1+(n-1)p_i)(1+(n-1)p_j)\big] \\
&\quad + n(n-1)p_i p_j p_j{}' \left[(n-2)[1+(n-3)p_i] - n(1+(n-1)p_i)\right] \\
&\quad + n(n-1)p_i p_j p_j{}'' \left[(n-2)[1+(n-3)p_i] - n(1+(n-1)p_i)\right] \\
&\quad + n(n-1)p_j p_i p_i{}' \left[(n-2)[1+(n-3)p_j] - n(1+(n-1)p_j)\right] \\
&\quad + n(n-1)p_j p_i p_i{}'' \left[(n-2)[1+(n-3)p_j] - n(1+(n-1)p_j)\right] \\
&\quad + n(n-1)p_i p_i{}' p_j p_j{}' \left[(n-2)(n-3) - n(n-1)\right] \\
&\quad + n(n-1)p_i p_i{}' p_j p_j{}'' \left[(n-2)(n-3) - n(n-1)\right] \\
&\quad + n(n-1)p_i p_i{}'' p_j p_j{}' \left[(n-2)(n-3) - n(n-1)\right] \\
&\quad + n(n-1)p_i p_i{}'' p_j p_j{}'' \left[(n-2)(n-3) - n(n-1)\right]\big) \\
&= n^{-3}p_i p_j \big[(n-1)\big(1+(n-2)(p_i + p_j + (n-3)p_i p_j) \\
&\qquad\qquad + (p_j{}' + p_j{}'')[(n-2)[1+(n-3)p_i] - n(1+(n-1)p_i)] \\
&\qquad\qquad + p_j p_i(p_i{}' + p_i{}'')[(n-2)[1+(n-3)p_j] - n(1+(n-1)p_j)] \\
&\qquad\qquad + (p_i{}' + p_i{}'')(p_j{}' + p_j{}'')[(n-2)(n-3) - n(n-1)]\big) \\
&\quad - n(1+(n-1)p_i)(1+(n-1)p_j)\big]
\end{aligned}
$$
(43)

Finally, we turn our attention to $\mathrm{Cov}[M_i, W_i]$, $\mathrm{Cov}[M_i, W_j]$, and $\mathrm{Cov}[W_i, M_j]$. Beginning with $\mathrm{Cov}[M_i, W_i]$:

$$
\begin{aligned}
\mathrm{Cov}[M_i, W_i] &= n^{-3}\mathrm{Cov}\left[N_i(N_i + N_i{}' + N_i{}''), N_i + N_i{}'\right] \\
&= n^{-3}\mathrm{Cov}\left[N_i^2 + N_i N_i{}' + N_i N_i{}'', N_i + N_i{}'\right] \\
&= n^{-3}\big(\mathrm{Cov}[N_i^2, N_i] + \mathrm{Cov}[N_i^2, N_i{}'] \\
&\qquad + \mathrm{Cov}[N_i N_i{}', N_i] + \mathrm{Cov}[N_i N_i{}', N_i{}'] + \mathrm{Cov}[N_i N_i{}'', N_i] + \mathrm{Cov}[N_i N_i{}'', N_i{}']\big).
\end{aligned}
$$

The terms in the expression above are given below. From (33):

$$
\begin{aligned}
\mathrm{Cov}[N_i^2, N_i] &= \mathrm{E}[N_i^3] - \mathrm{E}[N_i^2]\mathrm{E}[N_i] \\
&= np_i\left[1+(n-1)p_i(3+(n-2)p_i)\right] - np_i[1+(n-1)p_i]np_i \\
&= np_i\left[1+(n-1)p_i(3+(n-2)p_i) - np_i(1+(n-1)p_i)\right] \\
&= np_i\left[1 + p_i((n-1)[3-2p_i] - n)\right].
\end{aligned}
$$

From (38) and (33):

$$
\begin{aligned}
\mathrm{Cov}[N_i^2, N_i{}'] &= \mathrm{E}[N_i^2 N_i{}'] - \mathrm{E}[N_i^2]\mathrm{E}[N_i{}'] \\
&= n(n-1)p_i p_i{}'[1+(n-2)p_i] - np_i(1+(n-1)p_i)np_j \\
&= np_i p_i{}'\left[(n-1)(1+(n-2)p_i) - n(1+(n-1)p_i)\right].
\end{aligned}
$$
(44)

From (38) and (33):

$$\begin{aligned}
\mathrm{Cov}[N_i N_i{}', N_i] &= \mathrm{E}[N_i^2 N_i{}'] - \mathrm{E}[N_i N_i{}']\mathrm{E}[N_i] \\
&= n(n-1)p_i p_i{}'[1 + (n-2)p_i] - n(n-1)p_i p_i{}' n p_i \\
&= n(n-1)p_i p_i{}'[1 - 2p_i], \\
\mathrm{Cov}[N_i N_i{}', N_i{}'] &= n(n-1)p_i{}' p_i[1 - 2p_i{}'], \\
\mathrm{Cov}[N_i N_i{}'', N_i] &= n(n-1)p_i p_i{}''[1 - 2p_i].
\end{aligned}$$

From (39), (32), and (33):

$$\begin{aligned}
\mathrm{Cov}[N_i N_i{}'', N_i{}'] &= \mathrm{E}[N_i N_i{}'' N_i{}'] - \mathrm{E}[N_i N_i{}'']\mathrm{E}[N_i{}'] \\
&= n(n-1)(n-2)p_i p_i{}'' p_i{}' - n(n-1)p_i p_i{}'' n p_i{}' \tag{45} \\
&= -2n(n-1)p_i p_i{}'' p_i{}'.
\end{aligned}$$

Hence, $\mathrm{Cov}[M_i, W_i]$ is derived:

$$\begin{aligned}
\mathrm{Cov}[M_i, W_i] &= n^{-3}\big(n p_i[1 + p_i((n-1)[3 - 2p_i] - n)] \\
&\quad + n p_i p_i{}'[(n-1)(1 + (n-2)p_i) - n(1 + (n-1)p_i)] \\
&\quad + n(n-1)p_i p_i{}'[1 - 2p_i] \\
&\quad + n(n-1)p_i{}' p_i[1 - 2p_i{}'] \\
&\quad + n(n-1)p_i p_i{}''[1 - 2p_i] \\
&\quad - 2n(n-1)p_i p_i{}'' p_i{}'\big). \tag{46} \\
&= n^{-2}p_i\big(1 + p_i((n-1)[3 - 2p_i] - n) \\
&\quad + p_i{}'[(n-1)(1 + (n-2)p_i) - n(1 + (n-1)p_i)]\big) \\
&\quad + n^{-2}(n-1)(p_i p_i{}'[2 - 2p_i - 2p_i{}'] + p_i p_i{}''[1 - 2p_i - 2p_i{}']).
\end{aligned}$$

Then, moving on to $\mathrm{Cov}[M_i, W_j]$ and $\mathrm{Cov}[W_i, M_j]$:

$$\begin{aligned}
\mathrm{Cov}[M_i, W_j] &= n^{-3}\mathrm{Cov}\left[N_i(N_i + N_i{}' + N_i{}''), N_j + N_j{}'\right] \\
&= n^{-3}\mathrm{Cov}\left[N_i^2 + N_i N_i{}' + N_i N_i{}'', N_j + N_j{}'\right] \\
&= n^{-3}\big(\mathrm{Cov}[N_i^2, N_j] + \mathrm{Cov}[N_i^2, N_j{}'] \\
&\quad + \mathrm{Cov}[N_i N_i{}', N_j] + \mathrm{Cov}[N_i N_i{}', N_j{}'] + \mathrm{Cov}[N_i N_i{}'', N_j] + \mathrm{Cov}[N_i N_i{}'', N_j{}']\big).
\end{aligned}$$

The terms in the expression above are given below. From (44):

$$\begin{aligned}
\mathrm{Cov}[N_i^2, N_j] &= n p_i p_j\left[(n-1)(1 + (n-2)p_i) - n(1 + (n-1)p_i)\right], \\
\mathrm{Cov}[N_i^2, N_j{}'] &= n p_i p_j{}'\left[(n-1)(1 + (n-2)p_i) - n(1 + (n-1)p_i)\right].
\end{aligned}$$

From (45):

$$\begin{aligned}
\mathrm{Cov}[N_i N_i{}', N_j] &= -2n(n-1)p_i p_i{}' p_j, \\
\mathrm{Cov}[N_i N_i{}', N_j{}'] &= -2n(n-1)p_i p_i{}' p_j{}', \\
\mathrm{Cov}[N_i N_i{}'', N_j] &= -2n(n-1)p_i p_i{}'' p_j, \\
\mathrm{Cov}[N_i N_i{}'', N_j{}'] &= -2n(n-1)p_i p_i{}'' p_j{}'.
\end{aligned}$$

Hence, $\mathrm{Cov}[M_i, W_j]$ and $\mathrm{Cov}[W_i, M_j]$ are derived:

$$\begin{aligned}
\mathrm{Cov}[M_i, W_j] &= n^{-3}\big[n p_i(p_j + p_j{}')[(n-1)(1 + (n-2)p_i) - n(1 + (n-1)p_i)] \\
&\quad - 2n(n-1)p_i(p_i{}' p_j + p_i{}' p_j{}' + p_i{}'' p_j + p_i{}'' p_j{}')\big] \\
&= n^{-2}p_i(p_j + p_j{}')\big[(n-1)(1 + (n-2)p_i - 2(p_i{}' + p_i{}'')(p_j + p_j{}')) \\
&\quad - n(1 + (n-1)p_i)\big], \tag{47} \\
\mathrm{Cov}[W_i, M_j] &= n^{-2}p_j(p_i + p_i{}')\big[(n-1)(1 + (n-2)p_j - 2(p_j{}' + p_j{}'')(p_i + p_i{}')) \\
&\quad - n(1 + (n-1)p_j)\big].
\end{aligned}$$

Thus, all quantities necessary to compute (25) are derived.

