# OpenReview forum: "Bayesian Causal Bandits with Backdoor Adjustment Prior"
_TMLR — Accepted by TMLR_

### Review · Reviewer_E6Uy · 2022-11-25

**Summary Of Contributions:**

This work addresses causal bandits ---a multi-armed bandit setting where observable variables are modeled with a causal graph defining their conditional dependencies--- where the agent leverages the underlying causal relationships to more efficiently optimize its attained rewards.

The authors study causal bandits where data is observed as generated by the causal model, prior to the bandit agent playing (intervening on) any arms. Due to the dependencies inherent in the causal model, interventional distributions (i.e., those of the bandit's actions) may be inferred from observational distributions. Hence, information may be shared between arms for efficient exploration-exploitation strategies.

In contrast to existing work that assumes knowledge of the underlying causal graph, this work relaxes such assumption and combines causal graph learning with causal bandit interventions to devise a Bayesian bandit policy.

The proposed bandit algorithm computes inferences based on both observational (i.e., available prior to the bandit actions) and interventional (i.e., resulting from the agent's actions) data to simultaneously ($i$) learn about the underlying causal graph, and ($ii$) optimize the bandit's cumulative rewards, by exploiting causal inferences.

The proposed framework, named Bayesian Backdoor Bandit (BBB), quantifies the uncertainty in the expected reward estimates of the bandit agent by accounting for the reward signal and the unknown causal model, merging the following techniques:

1. Bayesian bandit policies, such as Bayes-UCB and Thompson sampling;
2. intervention calculus and the back-door formula, for estimating interventional quantities from observational data; and
3. Bayesian model averaging, to compute causal estimates that average over posterior distributions of plausible graphs, based on a technique proposed by Pensar et al. (2020).

Specifically, BBB implements a Bayesian bandit policy (Bayes-UCB or TS) with priors designed based on information from observational data (computed via the backdoor adjustment formula): the priors are initialized with expected value and variance estimates as presented in Equation (2). This prior is then sequentially updated to corresponding posteriors as the bandit plays (intervenes) on different arms.

Because the backdoor adjustment formula requires knowledge of the parents of a variable ---assumption not made by the authors here--- these estimates are computed by averaging over the posterior distribution of the plausible causal models, as indicated in Equations (3) and (4).

BBB leverages this carefully designed prior to execute Bayesian policies, where the posterior mean and variance statistics are updated as interventions are made, via averages over the posterior of the parent set distribution of Equation (4); i.e., the proposed Bayesian posterior incorporates uncertainties about both the reward function and the plausible causal graphs.

Key to the proposed BBB algorithm are the definition of the priors' mean and variance estimates, which the authors detail for two specific causal bandit applications: the discrete, multinomial bandit and the linear, Gaussian unit deviation setting. The derivation of the variance estimates for these two applications is the focus of Section 5 (with proofs provided in the Appendix), where approximations to these quantities of interest are provided. Given the proposed estimates' approximate nature, the authors focus on providing empirical evidence of the correctness and validity of these approximations, e.g., demonstrating coverage in Appendix Section C.

The work concludes with an evaluation section where the performance of BBB is compared to alternatives closest to this work (as per the discussion in Section 1.1) in a variety of simulated causal bandits. Results show that
1. BBB outperforms all non-causal counterpart policies, with performance monotonically improving with more observational data.
2. BBB achieves lower cumulative regret than the optimistic central node counterpart with enough (yet not excessive) observational data: $n_0 \geq 800$ in the discrete setting, $n_0 \geq 40$ in the Gaussian setting.

In addition, the authors showcase BBB's graph structure learning capabilities, describing the algorithm's performance tradeoff between the size of the observational data $n_0$ and the graph learning accuracy. With small observational data, initial uncertainty in both the structure prior and the reward encourages exploration of all arms, incurring in greater cumulative regret but allowing for identification of connections beyond the reward's parents; i.e., better graph recovery is possible than when BBB is initialized with bigger datasets ---where reduced initial uncertainty enables quick identification of the good arms, and thus small regret, yet the lack of exploration of other arms hinges learning the orientation of their edges.


**Audience:**

Yes

**Broader Impact Concerns:**

This work does not discuss broader impact concerns.

**Claims And Evidence:**

Yes

**Requested Changes:**

1. For completeness, and to help a reader not familiar with causal graph inference techniques, details on the following should be incorporated:
    1. Describing the algorithm developed by Pensar et al. (2020), which is used here to compute the parent set probabilities in Equation (4). This would facilitate reproducibility of BBB.
    2. Describing in detail the "standard assumptions of global and local parameter independence and parameter modularity" mentioned in Section 3.
    3. $\hat{SE}$ appears in Equation (2) without being previously defined. Given its importance ---the focus of Section 5 is on approximating it--- the authors should define it before its first appearance.

2. Evaluation section:
    1. Results are presented for bandits with $p=10$ variables and 3 parents for the reward variable. The authors should clarify the motivation behind these choices; e.g., are these numbers used due to computational constraints imposed by Pensar's posterior computation algorithm? In addition, the work would be strengthen with results showing how sensitive BBBs performance is to these choices.
    2. Empirical analysis/evidence showcasing the extra computational complexity of BBB is missing: how expensive is to compute parent set posteriors and averages over them?
    3. The provided figures, given that they are averages over realizations, should include error-bars illustrating performance variability.

3. The authors acknowledge that in this work, $X$ is a causally sufficient system with no unobserved confounders. A discussion on the challenges unobserved confounders raise, and how to leverage tools from the causal inference literature to mitigate or accommodate them would complement and strengthen this work.


**Strengths And Weaknesses:**

The presented work is solid, with the following strengths:

- The main contribution of this work is being able to bypass the restrictive assumption of knowing the true causal graph. In contrast to previous work by Lu et al. (2021), this work focuses on the non-asymptotic observational data regime: i.e., it does not rely on the large-sample observational setting of (Greenewald et al., 2019).

- The work is very clearly presented, with succinct arguments about this work's contributions (section 1.2) that contrast with related alternatives (section 1.1), before delving into necessary preliminaries (section 2) to design informative (causal-graph based) priors in Section 3 needed for their proposed Bayesian Backdoor Algorithms of Section 4.

- Section 5 contains details on how to compute mean and variance estimates for BBB's implementation for two specific use-cases.

- The authors empirically demonstrate how BBB compares to standard algorithms that assume knowledge of the causal structure. Importantly, they showcase that BBB outperforms all non-causal counterparts (with performance monotonically improving with more observational data) and that it achieves lower cumulative regret than the optimistic central node counterparts with enough observational data.

Some weaknesses of the presented work are:

- The increased computational cost of the proposed BBB algorithm (that requires computation of parent posteriors in Equation 4). Details on this computational overhead are not provided or discussed.

- BBB is a combination of existing techniques, i.e., the proposed method's significance is on merging them together.

- The work, which relies on informative priors and estimates learned from prior observational data, lacks a theoretical analysis on how such data impacts regret performance (this tradeoff is illustrated empirically). The authors leave the theoretical analysis as future work.

- The work assumes all variables in the graph are observed. A discussion on whether confounding could be accommodated via techniques from the causal inference literature would be of interest.

- BBB is instantiated for 2 specific models: a discrete bandit setting and a (linear) Gaussian Unit Deviation Setting. Details on the significance of the Gaussian unit-deviation setting, and a discussion on the challenges of other causal models would strengthen the paper.

---

> ### Author Response · Authors · 2022-12-19
> **Response to Reviewer E6Uy**
>
> Thank you for your feedback and comments. In our response, we shall attempt to sequentially answer your questions and requested changes along with any directly relevant weaknesses or limitations before discussing the remaining comments.
>
> Regarding describing the algorithm developed by Pensar et al. (2020), while their algorithm is useful for our purposes and we apply it in our main numerical experiments, we emphasize that the computation of (4) is not restricted to their algorithm. We have added simulation results using MCMC sampling to Section 6, and specified in Section 3 that we used both approaches in our empirical study. Hence, to avoid appearing unduly restrictive (especially due to the computational constraints of the exact algorithm), we prefer not to unnecessarily direct attention away from our proposed methodological framework. Regarding reproducibility, as stated in Appendix B, we have made the complete code for reproducing our results publicly available. In response to your comment, we  additionally note in Appendix B that we include instructions for reproducing our results as well. If strongly preferred, we are happy to include a brief summary of the algorithm in Appendix B in a revision.
>
> Regarding the "standard assumptions of global and local parameter independence and parameter modularity", we expanded that statement in Section 3 to clearly define these assumptions.
>
> Regarding $\hat{\mathrm{SE}}^2$, we added its definition immediately following (2) in Section 3.
>
> Regarding the motivations behind the choices of $p=10$ and 3 parents for the reward variable, we added two paragraphs to Appendix B for clarification. The limitation to $p=10$ was in the interest of extending the scope of our empirical investigation in other aspects, namely in representing a large number of random causal models and executing enough repetitions and time steps to reasonably assess the expected performance. The in-degree restriction was largely due to the difficulty in reliably generating random conditional probability distributions that have meaningful causal effects and reward signals.
>
> Regarding the sensitivity of the performance of BBB to the choices of $p$ and the number of parents, to our recollection, we did not observe any issues in cumulative regret performance to perturbations in these values.  The primary constraint is indeed computational because of our use of the algorithm in Pensar et al. (2020). Note that the efficiency of the parent set posterior algorithm efficiency is primarily affected by the specified maximum number of parents in the algorithm, not the maximum in-degree of the underlying structure. Furthermore, while increasing the specified maximum parent set size in the algorithm resulted in up to triple the runtime, it did not in general deteriorate cumulative regret performance.
>
> Instead of providing simulation comparisons assessing sensitivity to $p$ and maximum parents, we chose to focus our efforts on demonstrating the effectiveness of approximating the parent set posteriors with MCMC sampling in Section 6, which addresses the computational limitations imposed by either. If these comparisons are still desired, we are happy to do so in a revision.
>
> Regarding the extra computational complexity of BBB, we added a paragraph of discussion to Section 6, noting that the average execution time of each iteration of our discrete and Gaussian implementations of BBB on the random networks was respectively around 4.3 and 8.2 seconds, compared to less than 0.15 seconds for the competing methods.
>
> Regarding the error bars illustrating performance variability, we did not include error bars due to the density of information communicated in the figures, along with the substantial variability arising from the randomness in graph structures, conditional probability distributions, and data. Succinctly, the primary results and comparisons that we aim to communicate become impossible to discern when error bars are added. As such, we instead include a separate figure in Appendix C which visualizes the average cumulative regret with 95\% percentile error bars.
>
> Regarding unobserved confounders, we added a paragraph of discussion to Section 7. In the presence of unobserved confounders, an obvious challenge is that the variables in the parent sets that BBB uses for backdoor adjustment may not all be observed. Since sets that satisfy the backdoor criterion are not limited to the parent set, one approach would be to otherwise identify valid adjustment sets. Perhaps the most obvious extension of our methodology would be to model the underlying ancestral graph instead of the DAG. From the ancestral graph, causal effects may be estimated from observational data by identifying valid backdoor adjustment sets based on its structure; i.e., sets of ancestors that block backdoor paths, or by conducting the frontdoor adjustment instead of the backdoor adjustment.

---

> > ### Comment · Reviewer_E6Uy · 2022-12-27
> > **Thank you for your informative responses and revision**
> >
> > I thank the authors for a thorough revision of the manuscript that addresses our concerns and significantly improves it.
> >
> > Specifically, I thank them for addressing the computational aspects of BBB, by:
> >
> > - Clarifying that BBB's computation of graph posteriors in (4) is not restricted to any specific algorithm.
> > - The new empirical evaluation of BBB with both exact computation of parent set posteriors (via Pensar et al 2020) and approximations (via MCMC sampling).
> > - The acknowledgment that the computational requirements of BBB with exact parent set posterior computation clearly exceeds that of the alternatives.
> > - The additional implementation and results on using MCMC to approximate parent set posteriors for BBB, showing the scalability of BBB to larger problems in this case, and providing an informative discussion on the computational-performance tradeoff.
> >
> > In addition, the manuscript now contains
> >
> > - The outline of a preliminary cumulative regret analysis in Section 4.2.
> >
> > - Revised and additional figures showcasing the variability of the algorithms' performance.
> >
> > - A discussion in Section 7 on how, in the presence of unobserved confounders, alternative methodologies to identify the ancestral graph are required.
> >
> > - Informative and clarifying revisions that strengthen the exposition of their work.

---

> > > ### Comment · Reviewer_rebg · 2022-12-28
> > > **Corroborating the opinion of Reviewer E6Uy**
> > >
> > > The authors have claimed to address the majority of my concerns in the revision. I am appreciative of all these efforts. However, since the posted PDF does not contain blue highlights of the mentioned changes, it is difficult to verify whether these claims hold true. Can the action editor request two different versions of the manuscript be uploaded: the original version, and the one with revisions?

---

### Review · Reviewer_rebg · 2022-11-27

**Summary Of Contributions:**

This work develops the multi-armed bandit setting for the case that both observational and interventional data are available, and a directed acyclic graph relating them is available in the form of a prior that relates the causal structure of the problem. With this information in hand, a maximum a posteriori update rule for these probabilities is developed, and its uses in a way that can be incorporated as a prior into upper-confidence bound and Thompson sampling, is developed. Experimental analysis illuminates the merits of the proposed approach.

**Audience:**

Yes

**Claims And Evidence:**

Yes

**Requested Changes:**

In the discussion of limitations of assuming knowledge of the causal graph in prior work during Section 1.1, the authors should provide an illustrative example where this assumption breaks, so as to make clear that these conditions are in fact limiting. Moreover, the following paragraph discusses various efforts to relax this assumption and the limitations of those efforts as well; however, without specific technical reasons that their conditions are restrictive in the context of some problem in particular, the argument that these conditions are limiting comes across as hollow. Why is the estimation of interventional quantities from observational data more practical? is it because then the bandit algorithm does not depend on unverifiable assumptions? How are the conditions imposed by this approach weaker than the aforementioned ones?

In the summary of contributions, the authors should provide a more quantitative description of the main regret bound in the discrete, especially how it refines previous results' dependence on the prior, and how that prior exhibits specific dependence on the observational data which facilitates theoretical and potentially practical improvements. For the continuous space with Gaussian model, how does the proposed framework improve dependence on the prior, and what specific improvements are achieved through the use of interventional calculus? As written, these aspects are ambiguous.

In Section 3, it is mentioned that the class of priors is easier to evaluate when submodularity is present, due to the super-exponential growth with the dimension of the variables. However, is this referring to the dimension of the feature space or the action space? Moreover, which classes of priors actually satisfy this condition? How restrictive is it, and what are some practical examples?

The legend in Figures 1-2 seems incorrect. I think the red should refer to Bayes-UCB, and the green should refer to UCB, and the dotted to the backdoor adjustment prior variant. If it is correct, then it is ambiguous. Please correct the legend. What does Alg mean? I think the authors mean to contrast the performance of Thompson sampling/UCB with and without a backdoor adjustment prior.

References missing regarding the link between bandit algorithms and experimental design:

Krause, A., Singh, A., & Guestrin, C. (2008). Near-optimal sensor placements in Gaussian processes: Theory, efficient algorithms and empirical studies. Journal of Machine Learning Research, 9(2).

Bedi, A. S., Peddireddy, D., Aggarwal, V., & Koppel, A. (2020, July). Efficient large-scale gaussian process bandits by believing only informative actions. In Learning for Dynamics and Control (pp. 924-934). PMLR.

The authors should expend more effort to illuminate how the proposed backdoor adjustment prior improves dependencies on naive implementations of UCB or Thompson sampling for the proposed setting.

The analysis that gives rise to Proposition 1-2 does not appear novel in my reading. What is specifically different about these proofs as compared with the references from whence they came, and is there any attribute of the MAB setting or DAG structure associated with the causal inteventional setting that necessitates such a departure?

There are long run-on mathematical experiments in the appendices that no reader should be expected to parse. See equation (35) for an exmaple. These expressions should be broken up, and only each term that actually gets manipulated should be restated. One does not need to carry all the terms to explain what is going on.

**Strengths And Weaknesses:**

Strengths:

The problem definition in terms of introducing observational and interventional data as a prior into the multi-armed bandit framework appears novel, to the best of my knowledge.

The paper is well-written and defines the problem clearly.

Weaknesses:

The terminology if 'backdoor adjustment prior' is not commonplace, and relatively unexplained.

What is the quantitative difference between observational and interventional data? This is not well-defined, and therefore the intuition behind eqn. (1) is absent.

Similarly, interventional calculus is a term that is thrown around, but its meaning is not widely understood. Therefore, it should be more rigorously defined in the context of the introduction of the Bayesian prior.

To evaluate the backdoor adjustment prior in eqn. (3), we need access to the probabilities P(Pa_a = Z | D[t] ) which are associated with the DAG defining the causal structure. How reasonable is this assumption? Some practical instances when it is known, or not known, should be detailed.

A key weakness of this work is that there appears to be no regret bound analysis that establishes the performance of the proposed approach, and contrasts its theoretical rates with UCB or Thompson sampling without such knowledge of the casual DAG. Therefore, a rigorous understanding of when and why this approach outperforms some prior baselines is not very clear.

---

> ### Author Response · Authors · 2022-12-19
> **Response to Reviewer rebg**
>
> Thank you for your feedback and comments. In our response, we shall attempt to sequentially answer your questions and requested changes along with any directly relevant weaknesses or limitations before discussing the remaining comments.
>
> Regarding the quantitative difference between observational and interventional data, we revised Section 2 to provide a brief introduction to causal structural equation models to differentiate between the observational and interventional distributions. We also clarify our use of interventional calculus in reference to estimating the interventional distribution from observational quantities. We use the terminology of backdoor adjustment prior primarily descriptively for the formulation of BBB, which we clarify in the beginning of Section 3.
>
> Regarding the parent set probabilities in (3), we do not assume possession of these but compute them according to the posterior distribution of DAG structures based on the data, as seen in (4) and (5). In other words, we are always able to compute (or approximate) these parent set probabilities from data.
>
> Regarding the regret analysis, we added preliminary theoretical analysis in Section 4.2. In particular, we show that under certain assumptions, the Bayesian regret bound of BBB-UCB is $O(\sqrt{T \log T})$, independent of the number of arms $K$, when the observational data size $n_0$ is comparable to the total time steps $T$. This corroborates our empirical results, showing that the dependence of the regret on the number of actions can be relaxed by using observational data to simultaneously estimate $\mu_a$, $a \in \mathcal{A}$ using the backdoor adjustment. This bound is smaller than the regret bound $O(\sqrt{KT\log T})$ of standard MAB methods.  Further comparisons with other regret bounds of causal bandit methods can be made in a revision.
>
> Prior knowledge of the causal graph is a significantly limiting assumption. Intuitively, it is harder to imagine scenarios where the underlying causal structure is known than unknown. To facilitate understanding by way of illustration, we restated the farming example discussed in Lattimore et al. (2016) in Section 1.
>
> The super-exponential growth is with respect to the feature space. We clarify that modularity of the structure and parameter priors are standard assumptions in Bayesian network structure learning literature (Friedman \& Koller, 2003). Examples of modular structure priors include uniform over structures and over parent set sizes. Examples of modular parameter priors are Dirichlet for discrete networks and Wishart for Gaussian networks.
>
> We acknowledge that there is a lot of information being communicated in each of Figure 1 and Figure 2. For clarity as to Alg, we updated the subgraph labels to specify that Alg represents one of Bayes-UCB, TS, and UCB. However, we maintain that the information in the figures are correctly labeled. To further enhance clarity, we note that to circumvent confounding arising from the differences in algorithm designs and any relevant parameter tuning, we focus our comparisons within algorithm types, e.g. amongst TS, TS$^*$, and BBB-TS.
>
> We made revisions to the summary of contributions in Section 1.2 and the description of the backdoor adjustment prior in Section 3 to better distinguish BBB from standard non-causal bandit algorithms such as naive implementations of Bayes-UCB and Thompson Sampling. Summarized, whereas standard (Bayesian) algorithms assume non-informative priors, our proposed backdoor adjustment prior in BBB encodes causal inferences from finite samples of observational data to initialize informative expected reward posteriors. This framework of integrating of jointly observational and interventional data in BBB is applied in the discrete and Gaussian settings in Section 5.
>
> Regarding the proofs of Propositions 1 and 2, we do not claim novelty as to the theoretical analysis. Rather, the assertions in Propositions 1 and 2 are beneficial for the implementation of our methodology in the Gaussian case to maximally leverage observational data in constructing the informative priors and to share information between arms. We are not aware of instances where the propositions themselves are formalized elsewhere in literature, which we suspect may be due to their somewhat intuitive nature.
>
> Regarding the long run-on mathematical experiments, we attempted to include adequate details for completeness for the interested reader while simultaneously omitting extraneous and redundant details. We showed more detailed steps when establishing derivation strategies, but otherwise attempted to limit to stating an equation, manipulating if needed, plugging in the derived quantities, and showing 1-2 steps of simplification. Eq. (35) consists of only one step plugging in and one step attempting to simplify. We will attempt to simplify the derivation in a revision and are open to any specific suggestions on how we may better guide the reader through the derivation.

---

### Review · Reviewer_Veix · 2022-12-05

**Summary Of Contributions:**

This paper studies the causal bandits from the Bayesian perspective. Compared with prior works, it does not assume the known causal graph or some requirements in the causal graph class. Instead, it takes a Bayesian perspective by constructing the prior using observational data and updated the posterior using the interventional data. To achieve this, this paper design a new framework: Bayesian Backdoor Bandit (BBB) that simultaneously learns the causal graph and utilizing the causal inference to optimize the reward. To instantiate this idea, the paper discusses the implementations on both the discrete and gaussian setting. Empirically, it also shows superior performance over baseline algorithms, such as TS, UCB in small scale experiments.



**Audience:**

Yes

**Claims And Evidence:**

Yes

**Requested Changes:**

- It would be great to have the approximated version of BBB-TS, to showcase the more applicable version and how the approximation affects the overall performance.

- As pointed out above, it would be great to either have a regret analysis for the BBB framework, under either discrete or continuous setting; or it might be helpful to test the methods in more real-world datasets, or large scale datasets (apparently, the computational complexity might affects the feasibility of doing this).

- Section 3 is a little bit hard to follow, could we organize them utilizing paragraphs that suggest the main points?

- In the empirical section, it would be great to give a more detailed description on the structure setup from de Kroon et al. (2022).

**Strengths And Weaknesses:**

Strength:

- The assumptions of the proposed method is kind of weak, which makes it is closer to the application settings. Prior works mostly rely on known causal graphs, or the underlying causal graph is restrictive. This paper gets rid of them by taking a Bayesian perspective, however, it might worth to discussing that if we want to have some theoretical results for BBB, are the limited causal graph class is required?

- Most part of the paper is pretty well-written and easy to read. The methods and their instantiations are well-illustrated.

- The empirical section, though is based on synthetic data, shows some interesting observation, i.e., the larger sample size of observational data might even hurt the performance when the time step is longer. Is this also the case for other baseline algorithms?

Weakness:

- The main concern of the proposed method is the high computation cost in computing the parent set probabilities and the conditional parameter posteriors, which might limit its use and impact. Though the author discusses some approximation through MCMC might be possible for BBB-TS, it would be great to give a try and illustrate the robustness of the proposed method under the approximation setting.

- I would suggest the paper to dig into more investigations into either the theoretical or empirical side of the proposed method. For the current version of the paper, the regret analysis of the BBB framework is missing, and it would be great to understand more how it compared with the baselines theoretically. For the empirical side, it would be great to have some real-world datasets to illustrate the proposed BBB framework.

---

> ### Author Response · Authors · 2022-12-19
> **Response to Reviewer Veix**
>
> Thank you for your feedback and comments. In our response, we shall attempt to sequentially answer your questions and requested changes along with any directly relevant weaknesses or limitations before discussing the remaining comments.
>
> Regarding the high computational expense and the approximation with MCMC, we implemented an approximated version of BBB for the discrete case that uses structure MCMC. In particular, at each iteration, for BBB-TS a DAG is sampled with MCMC, and for BBB-(Bayes-)UCB the parent set posteriors are estimated by sampling DAGs with MCMC instead of exact computation. In an added subsection to Section 6, we discuss this implementation and provide empirical results. We first compare this approach to exact computation of parent set posteriors, showing that the approximation performs comparably with the exact approach. Note that it is difficult to empirically evaluate causal bandit algorithms using real-world datasets due to the need to perform interventions, so we evaluated the MCMC implementation of BBB on a larger realistic reference network with $p = 20$, again showing significant reductions in the cumulative regret compared to competing methods.
>
> Regarding the regret analysis, we added preliminary theoretical analysis in Section 4.2. In particular, we show that under certain assumptions, the Bayesian regret bound of BBB-UCB is $O(\sqrt{T \log T})$, independent of the number of arms $K$, when the observational data size $n_0$ is comparable to the total time steps $T$. This corroborates our empirical results, showing that the dependence of the regret on the number of actions can be relaxed by using observational data to simultaneously estimate $\mu_a$, $a \in \mathcal{A}$ using the backdoor adjustment. This bound is smaller than the regret bound $O(\sqrt{KT\log T})$ of standard MAB methods.  Further comparisons with other regret bounds of causal bandit methods can be made in a revision.
>
> Regarding the readability of Section 3, we organize the section under the headings of \textbf{Conditional Priors}, \textbf{Parent Set Averaging}, and \textbf{Structure Posterior}, correspondingly adding the following introductory sentence: "We begin by introducing the construction of conditional priors given backdoor adjustment sets before continuing to obtain the backdoor adjustment prior by Bayesian model averaging over parent set probabilities. We conclude by discussing the formulation and considerations of the posterior distribution over graph structures that gives rise to the parent set probabilities."
>
> Regarding the structure setup, we did indeed describe the random structure generation process in detail in Appendix B, but recognize that such a fact may not be apparent to the interested reader. As such, we clarify in Section 6 that comprehensive experimental details sufficient for reproducing our experiments, such as CBN preparation and algorithm parameters, are provided in Appendix B.

---

### Review · Reviewer_PzMw · 2022-12-07

**Summary Of Contributions:**

This study presents Bayes-flavored algorithms for the Causal bandit problem with frequent evaluation. As causal bandits require the knowledge of causal graphs, most existing studies develop their own algorithms given the information. In this context, the authors' methods incorporate the information as prior information of Bayes-flavored algorithms. Motivated by this, the authors develop several algorithms utilizing the properties of Bayesian algorithms. Furthermore, the author also simulation studies to demonstrate the superiority of the proposed method over existing methods.

**Audience:**

Yes

**Broader Impact Concerns:**

The proposed algorithms are easy to apply and implement. They are also helpful for practitioners.

**Claims And Evidence:**

Yes

**Requested Changes:**

I agree that the proposed framework of this study is practically useful and insightful in that it allows us to incorporate prior information into algorithms. However, (i) there are no theoretical results such as optimality and posterior convergence, and (ii) the notations are a bit hard to read, or the definitions (formulations) may have some issues. Because of these concerns, I believe that the formulation and theoretical analysis need to be further improved in order for this paper to be accepted.


**Strengths And Weaknesses:**

This study demonstrates a naive but powerful and natural approach to the causal bandit problem. Bayesian algorithms or algorithms motivated by Bayesian ideas can naturally incorporate prior information into algorithms. Among practitioners, employing Bayesian methods for incorporating prior knowledge is a well-known idea. Therefore, I believe that the main contribution lies in the confirmation of the effectiveness of this approach in the causal bandit problem. The proposed framework is innovative and intriguing because such algorithms have not been officially proposed in spite of their usefulness.

Although the proposed framework has some novelty, the lack of theoretical analysis makes this study’s evaluation difficult. To investigate the optimality of algorithms, it is preferable to show the lower and upper bounds for the proposed algorithms. For example, we can employ information-theoretic lower bounds of Lai and Robbins (1985) or lower bounds for an asymptotic Bayes evaluation to show the asymptotic optimality (Lai, 1987). If the authors consider a Bayesian evaluation instead of a frequentest evaluation, they might obtain Bays optimal algorithm by solving dynamic programming under appropriate settings on time horizon and discount rates, such as Gittins index (Gittins, 1989).


My main questions are listed as follows:
- Definitions of some notations, such as $E$, $\mathrm{Var}$, and $\hat{\mathrm{SE}}$ are desirable, although we can guess the meanings.
- On p.3, is $E[R_T] = T\mu^* 0 \sum_{a\in\mathcal{A}}$ is an expectation under posterior parameters? Similarly, is $\mu_a$ an expectation under a fixed parameter, and $E_{\pi^0_{a|\bm{Z}}}[\mu_a]$ is an expectation marginalized over the posterior distribution? For example, Kaufmann et al. (2012) denotes the cumulative regret by $R_n(\theta) = \mathbb{E}_\theta[\sum^n_{t=1}\mu^* - \mu_{I_t}]$ on Eq. (1) in their paper.
- What is $\hat{P}$ on (9). Is it a posterior probability given observations?
- Should $\mathrm{Var}(Y|dp(X=x))$ be denoted by $\mathrm{Var}(Y|dp(X=x), \pi^t_a)$ because it is the variance under a fixed parameter?
- Can the authors show the posterior convergence and consistency for some important parameters such as $\mu_a$ given the data generating process, even if they cannot show the optimality of their algorithms?
- In Appendix D.2, for $Q_i = f(M_i, W_i)$ the authors apply series expansion around $\mu_i = (\mu_{M_i}, \mu_{W_i})$. In Eq. (18), the authors omit the higher-order term of the second-order series expansion, but briefly writing them may be helpful for readers. Furthermore, I think that the higher-order term can be ignored only when the sample is infinite. Can the authors justify the use of approximation when the sample size is small?

1. Gittins, J. C. (1989): Multi-armed Bandit Allocation Indices. Wiley, Chichester, NY.
2. Lai, T., and H. Robbins (1985): “Asymptotically efficient adaptive allocation rules,” Advances in Applied Mathematics.
3. Lai, T. L. (1987): “Adaptive Treatment Allocation and the Multi-Armed Bandit Problem,” The Annals of Statistics, 15(3), 1091 – 1114.

---

> ### Author Response · Authors · 2022-12-19
> **Response to Reviewer PzMw**
>
> Thank you for your feedback and comments. In our response, we shall attempt to sequentially answer your questions and requested changes along with any directly relevant weaknesses or limitations before discussing the remaining comments.
>
>
> Regarding the questions:
>
> We added definitions of $\mathrm{E}$, $\mathrm{Var}$, and $\hat{\mathrm{SE}}$ with clarifications (in $P$ defined by $\mathcal{B}$, or in prior or posterior $\pi_a^t$) in the text for clarity.
>
> We have clarified the definitions of the regret and Bayesian regret in a new Section 4.2 on preliminary regret analysis in the revised paper.
>
> $\hat{P}$ is the empirical plug-in estimate of (1).
>
> Indeed, $\mathrm{Var}[Y \mid do(X = x)]$ is under probability distribution $P$ based on fixed CBN $\mathcal{B}$, so we have clarified it as $\mathrm{Var}_P$, which we defined in the text immediately following.
>
> We provided brief discussions as to the large-sample structure posterior behavior under Structure Identification in Section 6. From that, consistency of $\hat{\mu}_{a,\mathrm{bda}}$ follows from the consistency of the estimator (which is difficult to investigate in the discrete case) and consistency of correct parent set identification of node ${\langle a \rangle}$, the latter of which is straightforward in the infinite horizon setting. If desired, we are happy to include extended discussion in a revision.
>
> We added the higher order term $O \left( \lvert\lvert(M_i, W_i) - \mu_i \rvert\rvert^3 \right)$. In Appendix C.1, we conducted a simulation study demonstrating good finite-sample confidence interval coverage performance of our proposed estimator with modest sample sizes as low as $n_0 = 100$.
>
>
> Regarding the regret analysis, we added preliminary theoretical analysis in Section 4.2. In particular, we show that under certain assumptions, the Bayesian regret bound of BBB-UCB is $O(\sqrt{T \log T})$, independent of the number of arms $K$, when the observational data size $n_0$ is comparable to the total time steps $T$. This corroborates our empirical results, showing that the dependence of the regret on the number of actions can be relaxed by using observational data to simultaneously estimate $\mu_a$, $a \in \mathcal{A}$ using the backdoor adjustment. This bound is smaller than the regret bound $O(\sqrt{KT\log T})$ of standard MAB methods.  Further comparisons with other regret bounds of causal bandit methods can be made in a revision.

---

### Decision · Action_Editors · 2023-01-09

**Recommendation:** Accept as is

**Comment:**

This paper studies a Bayesian variant of causal bandits. The advantage of the Bayesian framework is that the causal graph does not have to be specified in advance and is learned on the go by maintaining its posterior. The authors evaluate their approach empirically and also derive a preliminary regret bound.

The initial version of this paper was already solid. In the rebuttal period, the authors comprehensively addressed the comments of the reviewers. They also went beyond the original writeup in two major ways:

* A preliminary regret analysis. The initial writeup had none.

* An approximate MCMC implementation of the algorithm. This is one major strength of the Bayesian view, that many approximations (although not with regret guarantees) exist.

I suggest acceptance of this paper **AS IS**, with the understanding that the authors expand related work (Section 1.1) to discuss Bayesian bandits better. There are many recent works that show algorithmic and regret improvements due to a better prior. Here are some pointers to consider:

* Modern Bayesian analyses of TS and Bayes-UCB (precursor to showing improvements due to the prior) started in [Learning to Optimize via Posterior Sampling](https://pubsonline.informs.org/doi/abs/10.1287/moor.2014.0650).

* An actual improvement is shown in [Information-Theoretic Confidence Bounds for Reinforcement Learning](https://proceedings.neurips.cc/paper/2019/hash/411ae1bf081d1674ca6091f8c59a266f-Abstract.html). A tighter MAB analysis, which is also easier to follow, is in Lemma 4 of [Meta-Thompson Sampling](https://proceedings.mlr.press/v139/kveton21a.html).

* Bandit meta-learning, essentially learning the prior for TS by repeatedly solving similar bandit tasks, received a lot of attention recently:

  * [Meta Dynamic Pricing: Transfer Learning Across Experiments](https://papers.ssrn.com/sol3/papers.cfm?abstract_id=3334629)

  * [No Regrets for Learning the Prior in Bandits](https://proceedings.neurips.cc/paper/2021/hash/ec1f764517b7ffb52057af6df18142b7-Abstract.html)

  * [Bayesian Decision-Making Under Misspecified Priors with Applications to Meta-Learning](https://proceedings.neurips.cc/paper/2021/hash/ddcbe25988981920c872c1787382f04d-Abstract.html)

  * [Metadata-Based Multi-Task Bandits with Bayesian Hierarchical Models](https://proceedings.neurips.cc/paper/2021/hash/f7cfdde9db36af8e0d9a6d123d5c385e-Abstract.html)

  * [Metalearning Linear Bandits by Prior Update](https://proceedings.mlr.press/v151/peleg22a.html)

* Fully Bayesian methods, where the prior reflects the structure of the problem, have been especially successful:

  * Posterior sampling for choosing one model out of many ([Latent Bandits Revisited](https://proceedings.neurips.cc/paper/2020/hash/9b7c8d13e4b2f08895fb7bcead930b46-Abstract.html))

  * Posterior sampling with a mixture prior ([Thompson Sampling with a Mixture Prior](https://proceedings.mlr.press/v151/hong22b.html))

  * Posterior sampling for meta-, multi-task, and federated learning ([Hierarchical Bayesian Bandits](https://proceedings.mlr.press/v151/hong22c.html))

Congratulations to acceptance!

**Audience:**

Yes. This paper is on adaptive learning (bandits) with a Bayesian prior (over the causal graph). This combination is common in the bandit literature and has practical solutions based on posterior sampling.

**Claims And Evidence:**

Yes. This paper supports its claims with both a regret bound and empirical results.